# PJA1-mediated suppression of pyroptosis as a driver of docetaxel resistance in nasopharyngeal carcinoma

Sheng-Yan Huang[1,2], Sha Gong[1,2], Yin Zhao [1,2], Ming-Liang Ye[1,2], Jun-Yan Li [1], Qing-Mei He[1], Han Qiao[1], Xi-Rong Tan[1], Jing-Yun Wang[1], Ye-Lin Liang[1], Sai-Wei Huang[1], Shi-Wei He[1], Ying-Qin Li [1], Sha Xu [1], Ying-Qing Li [1] ✉ & Na Liu [1] ✉

Chemoresistance is a main reason for treatment failure in patients with nasopharyngeal carcinoma, but the exact regulatory mechanism underlying chemoresistance in nasopharyngeal carcinoma remains to be elucidated. Here, we identify PJA1 as a key E3 ubiquitin ligase involved in nasopharyngeal carcinoma chemoresistance that is highly expressed in nasopharyngeal carcinoma patients with nonresponse to docetaxel-cisplatin-5-fluorouracil induction chemotherapy. We find that PJA1 facilitates docetaxel resistance by inhibiting GSDME-mediated pyroptosis in nasopharyngeal carcinoma cells. Mechanistically, PJA1 promotes the degradation of the mitochondrial protein PGAM5 by increasing its K48-linked ubiquitination at K88, which further facilitates DRP1 phosphorylation at S637 and reduced mitochondrial reactive oxygen species production, resulting in suppression of GSDME-mediated pyroptosis and the antitumour immune response. PGAM5 knockdown fully restores the docetaxel sensitization effect of PJA1 knockdown. Moreover, pharmacological targeting of PJA1 with the small molecule inhibitor RTA402 enhances the docetaxel sensitivity of nasopharyngeal carcinoma in vitro and in vivo. Clinically, high PJA1 expression indicates inferior survival and poor clinical efficacy of TPF IC in nasopharyngeal carcinoma patients. Our study emphasizes the essential role of E3 ligases in regulating chemoresistance and provides therapeutic strategies for nasopharyngeal carcinoma based on targeting the ubiquitin-proteasome system.

Nasopharyngeal carcinoma (NPC) is highly prevalent in South China, Southeast Asia and North Africa[1,2]. More than 70% of patients are diagnosed with locoregionally advanced NPC (LA-NPC) at initial presentation, and the addition of induction chemotherapy (IC) to concurrent chemoradiotherapy is recommended for these patients[3,4]. Our previous phase 3 randomised clinical trial suggests that docetaxel-cisplatin-5-fluorouracil (TPF) is an effective IC regimen[5,6]. However, the tumour response to TPF IC differs among patients, and a subgroup of patients who do not respond to TPF IC derive little benefit from IC and eventually experience recurrence or distant metastasis because of chemoresistance[7,8]. The exact regulatory mechanisms underlying NPC chemoresistance remain unclear and warrant further investigation.

[1]State Key Laboratory of Oncology in South China, Guangdong Key Laboratory of Nasopharyngeal Carcinoma Diagnosis and Therapy, Guangdong Provincial Clinical Research Center for Cancer, Sun Yat-sen University Cancer Center, Guangzhou 510060, P.R. China. [2]These authors contributed equally: Sheng-Yan Huang, Sha Gong, Yin Zhao, Ming-Liang Ye. ✉e-mail: liyingq1@sysucc.org.cn; liun1@sysucc.org.cn

Pyroptosis is defined as gasdermin (GSDM)-dependent programmed necrotic cell death, and it is triggered by various extracellular stimuli, such as viruses, chemotherapeutic agents, and nanoparticles[9–11]. Proteolytic cleavage of GSDM by active caspases liberates its N-terminal (NT) fragment to form pores in the plasma membrane and triggers the release of inflammatory cytokines (e.g. IL-1β and IL18) and damage-associated molecular patterns (DAMPs) into the extracellular space, resulting in activation of the immune system[9,12]. Recently, GSDME-mediated pyroptosis is reported to stimulate antitumour immunity and inhibit tumour growth[13,14]. In tumours with high GSDME expression, GSDME can induce a switch from caspase-3-mediated apoptosis induced by chemotherapeutic agents to pyroptosis[10]. In GSDME-deficient melanoma, targeted inhibitors fail to promote cleavage of GSDME and release of HMGB1, leading to a reduction in the infiltration of activated dendritic cells (DCs) and tumour-associated T cells[15]. However, the role of chemotherapy-induced pyroptosis in NPC and whether it can activate antitumour immunity are largely unknown.

The ubiquitin-proteasome system (UPS) is responsible for the degradation of misfolded, damaged, or unneeded cellular proteins under both normal homoeostatic and stress conditions, and it regulates almost every cellular process[16,17]. As key regulators of the system, E3 ubiquitin ligases play vital roles in the development, progression and therapeutic response of various cancers; thus, they are desirable drug targets[18,19]. Our previous studies identified TRIM21 and TRIM25 as key radiotherapy-related E3 ligases and demonstrated their roles in regulating radioresistance and radiotherapy-induced antitumour immunity in NPC[20,21]. However, little is known about the roles of E3 ligases in regulating chemoresistance in NPC. PRAJA ring finger ubiquitin ligase 1 (PJA1) is an E3 ubiquitin ligase belonging to the PRAJA protein family[22]. PJA1 has been reported to promote tumorigenesis and metastasis by destabilising several tumour suppressor proteins, such as ELF, CIC and pSMAD3[22–24]. However, the role of PJA1 in NPC and whether and how it affects chemoresistance remain unknown.

Here, by analysing microarray data, we identify PJA1 as a key E3 ligase involved in chemoresistance in NPC that is highly expressed in patients who do not respond to TPF IC. PJA1 promotes the ubiquitination and degradation of the mitochondrial protein PGAM5, further facilitating the phosphorylation of DRP1 and reducing the production of reactive oxygen species (ROS) in mitochondria, in turn suppressing GSDME-mediated pyroptosis and the antitumour immune response. Pharmacological targeting of PJA1 with the small molecule inhibitor RTA402 enhances the docetaxel sensitivity of NPC. Clinically, high PJA1 expression indicates inferior survival and poor clinical efficacy of TPF IC in NPC patients. Our study emphasises the essential role of E3 ligases in regulating chemoresistance in NPC and provides therapeutic strategies for NPC based on targeting the UPS.

## Results

### PJA1 facilitates docetaxel resistance in NPC

Through analysis of a genome-wide mRNA expression profiling dataset (GSE132112)[8], we identified 385 differentially expressed chemosensitivity-related genes between NPC patients with response or nonresponse to TPF IC (Fig. 1a). Seven gene modules containing 30 core genes were further identified through analysis of the STRING database with the Cytoscape MCODE plugin, which revealed a link between these modules and the efficacy of TPF IC. The E3 ubiquitin ligase module was the top-ranked module, suggesting that dysregulation of the UPS might play an important role in chemoresistance in NPC. Since PJA1 was located in the centre of this module (Fig. 1b and Supplementary Fig. 1a), we selected it for further analysis. PJA1 was highly expressed in NPC patients with nonresponse to TPF IC (Fig. 1c), suggesting that PJA1 might function as a key E3 ligase and confer TPF chemoresistance in NPC.

To determine the clinical significance of PJA1 in NPC patients, we further conducted IHC staining of 279 paraffin-embedded NPC tissue samples with an antibody against PJA1. We observed positive expression of PJA1 in both the nucleus and cytoplasm and divided the tissue samples into four groups based on the staining intensity (strong, moderate, weak, or negative) (Supplementary Fig. 1b). We then divided the corresponding NPC patients into high and low PJA1 expression groups for Kaplan–Meier survival analysis and found that patients with high PJA1 expression had significantly worse disease-free survival (DFS), overall survival (OS), and distant metastasis-free survival (DMFS) (Supplementary Fig. 1c–e). Importantly, these results further demonstrated that patients with low PJA1 expression could benefit from TPF IC in terms of better DFS, OS and DMFS, while patients with high PJA1 expression did not benefit from the TPF IC (Fig. 1d and Supplementary Fig. 1f–i). Collectively, our data suggest that high PJA1 expression predicts poor TPF IC efficacy in NPC patients.

To gain insight into the biological role of PJA1 in TPF chemoresistance, we transiently knocked down PJA1 expression in HONE1 and SUNE1 cells (Supplementary Fig. 2a, b) and exposed these cells to a series of concentrations of docetaxel, cisplatin or 5-fluorouracil. The results of Cell Counting Kit-8 (CCK8) assays revealed that knockdown of PJA1 significantly increased the sensitivity of NPC cells to docetaxel (Fig. 1e) but not to cisplatin or 5-fluorouracil (Supplementary Fig. 2c, d). Meanwhile, knockdown of PJA1 inhibited the growth of NPC cells (Supplementary Fig. 2e). Then, we established a docetaxel-resistant SUNE1 cell line (named S-DR) by gradually adding the concentration of docetaxel (Supplementary Fig. 2f), and found that the IC50 value of docetaxel in S-DR cells was higher than that in the corresponding parental cells and knockdown of PJA1 increased the docetaxel sensitivity of S-DR cells (Supplementary Fig. 2g). Moreover, knockdown of PJA1 in NPC cells and S-DR cells significantly promoted cell death after treatment with docetaxel (Fig. 1f and Supplementary Fig. 2h).

We then injected SUNE1 cells with or without stable PJA1 knockdown into nude mice and then treated the mice with docetaxel to establish a xenograft model (Fig. 1g). The tumours in the PJA1-knockdown group were smaller and weighed less than those in the control group, especially after docetaxel administration, indicating that the tumours in the PJA1-knockdown group were more sensitive to docetaxel (Fig. 1h–I and Supplementary Fig. 2i). Taken together, these data indicate that PJA1 facilitates docetaxel resistance in NPC.

### PJA1 suppresses docetaxel-induced pyroptosis

Interestingly, we observed characteristic morphological features of pyroptosis, manifested as cell swelling and large plasma membrane blebs, in NPC cells treated with docetaxel (Supplementary Fig. 2j). Since it has been reported that chemotherapy drugs, such as cisplatin, doxorubicin and etoposide, can induce pyroptosis through caspase-3-mediated GSDME cleavage in tumour cells with high expression of GSDME[10,25], we hypothesised that PJA1 may suppress docetaxel-induced pyroptosis to facilitate docetaxel resistance. As expected, we found high expression of GSMDE in HONE1 and SUNE1 cells compared with GSDME-negative MDA-MB-468 cells (Supplementary Fig. 2k). We further confirmed that the activation of caspase-3 and the cleavage of GSDME were induced by docetaxel in a time- and dose-dependent manner (Supplementary Fig. 2l, m). Moreover, lactate dehydrogenase (LDH) release was significantly increased after exposure to docetaxel (Supplementary Fig. 2n), demonstrating that docetaxel treatment induces pyroptosis in NPC cells.

We then investigated whether PJA1 regulates docetaxel-induced pyroptosis in NPC cells. Knockdown of PJA1 promoted docetaxel-induced pyroptosis, as evidenced by the more obvious appearance of morphological features of pyroptosis as well as increased LDH release, caspase-3 activation and GSDME cleavage (Fig. 1j–l). In addition, flow cytometric analysis showed that knockdown of PJA1 increased the

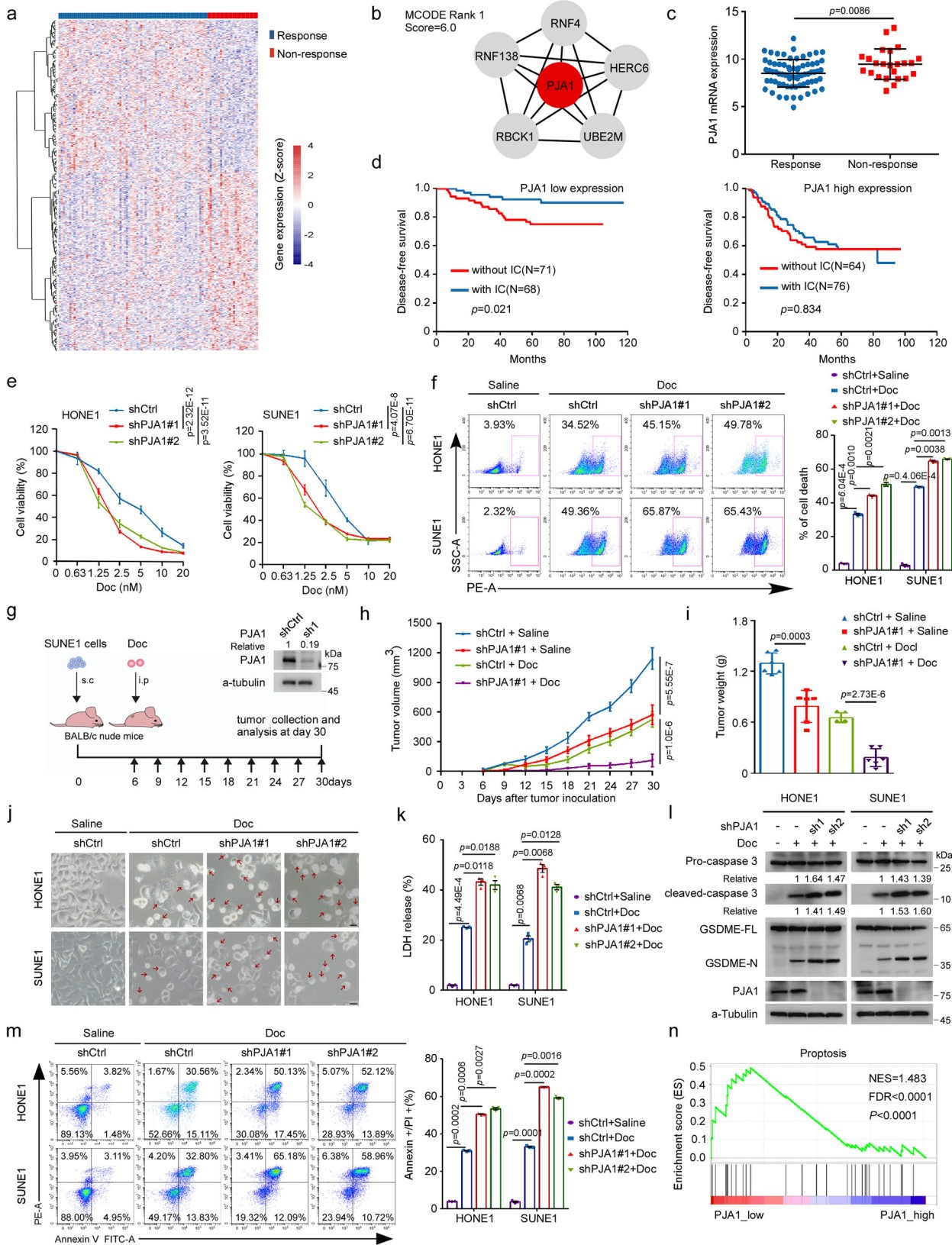

proportion of annexin V$^+$/PI$^+$ cells (Fig. 1m). Moreover, gene set enrichment analysis (GSEA) with one specific pyroptosis gene set consisting of 20 genes[26] revealed that the expression of most pyroptosis-related genes was decreased in the high PJA1-expression group (Fig. 1n). These results indicate that PJA1 can suppress docetaxel-induced pyroptosis to facilitate docetaxel resistance in NPC.

## PJA1 promotes the degradation of PGAM5 by increasing its K48-linked ubiquitination

To elucidate the molecular mechanism by which PJA1 suppresses docetaxel-induced pyroptosis, we performed mass spectrometry analysis and identified PGAM5, which has been reported to modulate mitochondrial dynamics[27–30], as a potential target of PJA1 (Fig. 2a,

**Fig. 1 | PJA1 facilitates docetaxel resistance by repressing pyroptosis in NPC.**
**a**–**c** Heatmap showing differentially expressed chemosensitivity-related genes (fold-change ≥1.5, q value <0.05) (**a**), Module analysis of the protein–protein interaction (PPI) network (**b**), and analysis of PJA1 expression (**c**, mean ± s.d., two-tailed unpaired t-test) in NPC tumours from patients with a response (n = 71) or nonresponse (n = 24) to induction chemotherapy (IC) with the docetaxel-cisplatin-5-fluorouracil (TPF) regimen, based on microarray data (GSE132112)[8].
**d** Kaplan–Meier analysis of disease-free survival in NPC patients treated with (n = 68) or without (n = 71) IC in the low PJA1 expression, and Kaplan–Meier analysis of disease-free survival in NPC patients treated with (n = 76) or without (n = 64) IC in the high PJA1 expression groups (log-rank test). **e** CCK8 assay measuring the chemosensitivity of NPC cells transfected with the shCtrl or sh-PJA1s plasmids and exposed to the indicated concentrations of docetaxel (Doc, mean ± s.d., two-way ANOVA). **f** Flow cytometry analysis of cell death in NPC cells transfected with the shCtrl or sh-PJA1s plasmids and exposed to Doc (10 nM) for 48 h (mean ± s.d., one-

way ANOVA). **g**–**i** SUNE1 cells stably transfected with the shCtrl or sh-PJA1 plasmids were implanted subcutaneously into BALB/c nude mice to establish a xenograft growth model, and the mice were treated with or without Doc (10 mg/kg) (**g**). Growth curves (**h**, two-tailed Student's t-test) and weights of the excised tumours (**i**, two-tailed unpaired t-test) in each group (mean (n = 6) ± s.d.). **j**–**m** NPC cells were transfected with the shCtrl or sh-PJA1 plasmids and exposed to Doc (10 nM). Representative images of pyroptotic morphology. The red arrow indicates pyroptotic cells. Scale bar, 25 μm. **j** LDH release (**k**, mean ± s.d., one-way ANOVA), caspase-3 and GSDME cleavage (**l**) and the proportion of annexin V+/PI+ cells (**m**, mean ± s.d., one-way ANOVA) were shown. **n** Gene set enrichment analysis (GSEA) indicated that the pyroptosis pathway was enriched in the low PJA1-expression group (GSE132112)[8] (permutation test). One-way ANOVA with Dunnett's multiple comparisons test; n = 4 (**e**), n = 3 (**f**, **j**, **k**, **m**) repeats from three independent experiments. Source data are provided as a Source Data file.

Supplementary Fig. 3a and Supplementary Data 1). The exogenous and endogenous interactions between PJA1 and PGAM5 were verified by coimmunoprecipitation (Co-IP), and their interactions were strengthened upon docetaxel treatment (Fig. 2b and Supplementary Fig. 3b, c). Additionally, this interaction was verified by the colocalization of PJA1 and PGAM5 in mitochondria, as shown by immunofluorescence (IF) staining (Fig. 2c).

As PJA1 is an E3 ubiquitin ligase, we sought to determine whether it affects the protein stability of PGAM5. Overexpression of PJA1 decreased both the exogenous and endogenous protein expression of PGAM5 in a dose-dependent manner, while knockdown of PJA1 increased the endogenous protein expression of PGAM5 but did not affect its mRNA expression level (Fig. 2d, e and Supplementary Fig. 3d). Consistent with the above results, overexpression of PJA1 significantly promoted but knockdown of PJA1 significantly suppressed the degradation of both exogenous and endogenous PGAM5 after treatment with cycloheximide (CHX) (Fig. 2f and Supplementary Fig. 3e), suggesting that PJA1 shortens the half-life of the PGAM5 protein. In addition, the protein expression of PGAM5 was increased in the tumours of mice in the PJA1-knockdown group than those in the control group, as determined by immunohistochemical (IHC) staining (Supplementary Fig. 3f). Furthermore, PJA1 protein expression was increased while PGAM5 protein expression was decreased in NPC tumours with nonresponse to TPF IC than those with response (Supplementary Fig. 3g). These results demonstrate that PJA1 promotes the degradation of the PGAM5 protein in NPC cells.

To determine whether PJA1 promotes PGAM5 degradation mediated by the ubiquitin-proteasome pathway or by the autophagy-lysosomal pathway, HEK293T cells were treated with MG132 (a proteasome inhibitor) or chloroquine (CQ; a lysosome inhibitor) after co-transfection with Flag-PJA1 and HA-PGAM5. PJA1-mediated degradation of PGAM5 was reversed by MG132 but not by CQ (Supplementary Fig. 3h). In addition, the degradation of endogenous PGAM5 mediated by PJA1 was reversed by MG132 but not by CQ in NPC cells (Fig. 2g, h), confirming that PJA1 promotes PGAM5 degradation through the ubiquitin-proteasome pathway. We then examined the effect of PJA1 on the ubiquitination of PGAM5 and found that overexpression of PJA1 increased WT- and K48-linked but not K63-linked polyubiquitination of PGAM5, which was strengthened upon docetaxel treatment (Fig. 2i and Supplementary Fig. 3i). Conversely, knockdown of PJA1 decreased WT- and K48-linked polyubiquitination of PGAM5 (Fig. 2j). Our results reveal that PJA1 promotes the degradation of PGAM5 by increasing its K48-linked polyubiquitination.

## PJA1 catalyses PGAM5 polyubiquitination at K88 depending on its E3 ligase activity

A previous study reported that mutation of the conserved cysteine residue within the PJA1 RING finger domain in mouse transcripts to alanine (PJA1-C353A) leads to loss of its E3 ligase function[31]. Through

sequence alignment, we found that the conserved cysteine residue was cysteine 598 (C598) in human transcripts (Fig. 3a). We subsequently constructed two ligase-dead mutants of PJA1, a RING domain-deletion mutant (ΔRING) and a catalytically inactive mutant (C598A) (Fig. 3a). We found that both mutants lost the ability to degrade exogenous and endogenous PGAM5 proteins (Fig. 3b, c). Consistent with this finding, overexpression of PJA1-WT promoted the polyubiquitination of PGAM5, while overexpression of the ΔRING or C598A mutant did not (Fig. 3d), indicating that PJA1 ubiquitinates and degrades PGAM5 in a manner dependent on its E3 ligase activity.

To further identify the potential ubiquitinated lysine(s) of PGAM5, we performed mass spectrometry analysis and found that the lysine 88 (K88) in the PGAM5 protein contained a diglycine modification (Fig. 3e, f and Supplementary Table 1). We then generated a K/R mutant of PGAM5 (PGAM5-K88R) and found that the mutant was resistant to the PJA1-mediated polyubiquitination and degradation (Fig. 3g, h). Consistent with these findings, the half-life of PGAM5-K88R was prolonged compared with that of PGAM5-WT (Fig. 3i). These results indicate that PJA1 ubiquitinates PGAM5 at K88 in a manner dependent on its E3 ligase activity and thus promotes PGAM5 degradation.

## PJA1 inhibits docetaxel-induced mitochondrial damage via the PGAM5-DRP1 axis

A previous report indicated that PGAM5 can recruit the mitochondrial fission factor DRP1 and dephosphorylate it at serine 637 (S637), thus promoting mitochondrial fragmentation and ROS production (Fig. 4a)[27]. Thus, we first examined the interactions among the PGAM5, DRP1 and PJA1 proteins. The results verified the exogenous interaction between PGAM5 and DRP1 and their colocalization in mitochondria (Fig. 4b and Supplementary Fig. 4a). Moreover, we confirmed the endogenous interactions of DRP1 with PJA1 and PGAM5 (Fig. 4c). Next, we examined whether PJA1 affects the phosphorylation of DRP1 and found that knockdown of PJA1 inhibited but overexpression of PJA1 promoted the phosphorylation of DRP1 at S637 (Fig. 4d and Supplementary Fig. 4b). Then, we found that overexpression of PGAM5-WT inhibited the phosphorylation of DRP1, and this effect was more obvious when PGAM5-K88R was overexpressed and was further potentiated by knockdown of PJA1 (Fig. 4e and Supplementary Fig. 4c). These findings demonstrate that PJA1 can promote the phosphorylation of DRP1 through ubiquitination of PGAM5.

As phosphorylation of DRP1 is essential for mediating peripheral division under stress conditions and is a marker of mitochondrial damage[32,33], we then tested whether PJA1 regulates docetaxel-induced mitochondrial damage. Confocal microscopy demonstrated that knockdown of PJA1 decreased the proportion of tubular and intermediate mitochondria but increased the proportion of fragmented mitochondria (Fig. 4f). Moreover, knockdown of PJA1 significantly reduced the mitochondrial membrane potential (Fig. 4g). Since mitochondrial damage can lead to metabolic dysfunction, manifesting as

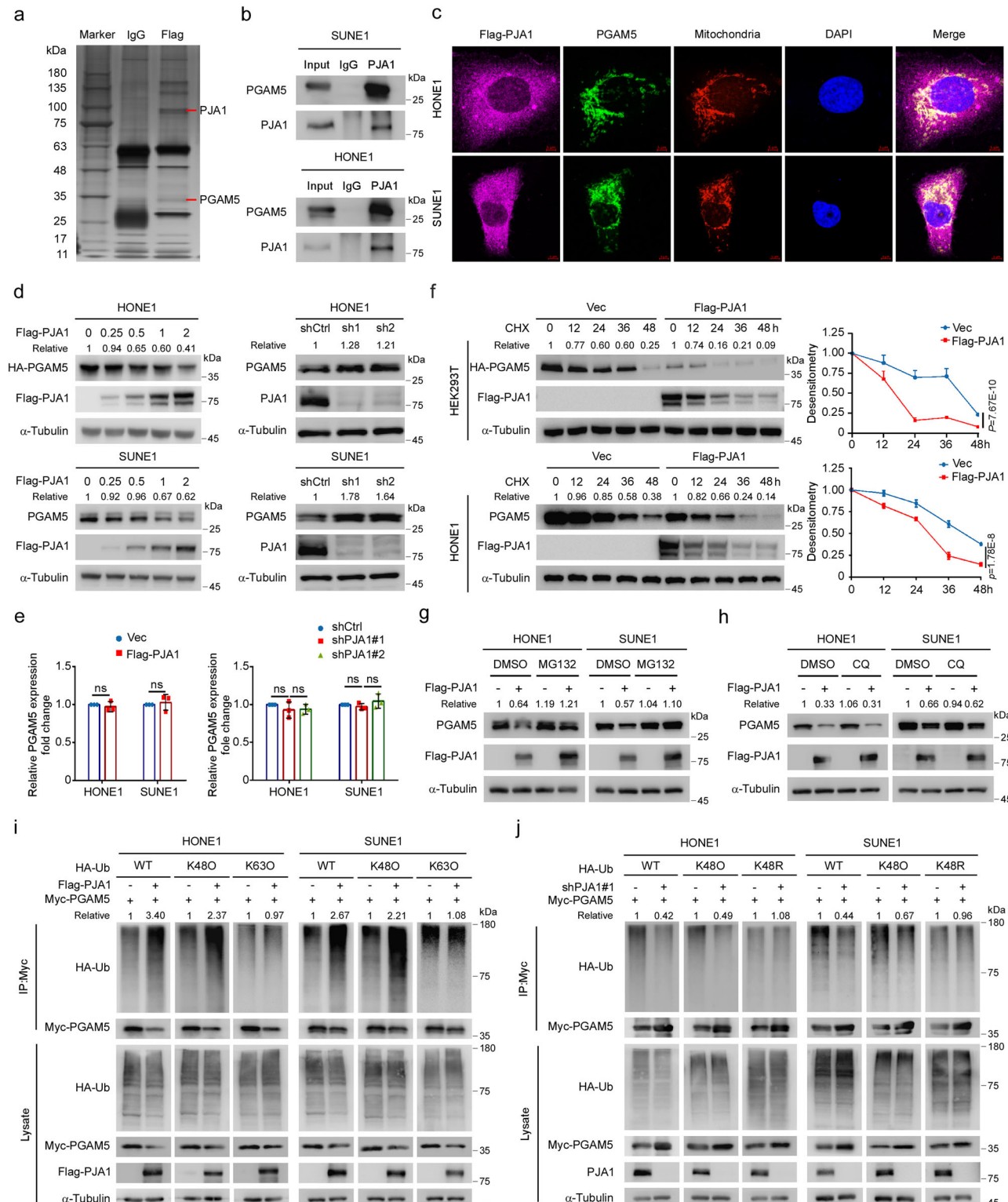

**Fig. 2 | PJA1 promotes PGAM5 degradation by increasing its K48-linked ubiquitination. a** Silver staining of SDS–PAGE gels showed that the Flag-immunoprecipitants were pulled down from SUNE1 cells overexpressing Flag-PJA1. Red lines indicate the proteins of interest. **b** Co-IP with an anti-PJA1 antibody revealed the endogenous association of PJA1 and PGAM5 in NPC cells. **c** IF staining revealed the cellular localisation of exogenous Flag-PJA1 (purple), endogenous PGAM5 (green) and mitochondria (red) in NPC cells. Scale bars, 5 μm. **d, e** Protein (**d**) and mRNA (**e**, mean ± s.d., two-tailed unpaired $t$-test (left), one-way ANOVA with Dunnett's multiple comparisons tests (right)) expression of PGAM5 in NPC cells transfected with gradient concentrations of the Flag-PJA1 plasmids, as well as in NPC cells transfected with the shCtrl or sh-PJA1s plasmids. **f** Immunoblot (left) and

the corresponding greyscale analysis (right) of PGAM5 expression in HEK293T and HONE1 cells transfected with the HA-PGAM5 plasmids together with the empty vector or Flag-PJA1 plasmids after the CHX treatment (mean ± s.d., two-way ANOVA). **g, h** PGAM5 protein levels in NPC cells transfected with the empty vector or Flag-PJA1 plasmids after the treatment with MG132 (**g**) or CQ (**h**). **i, j** NPC cells transfected with the empty vector or Flag-PJA1 plasmids (**i**) or with the shCtrl or sh-PJA1 plasmids (**j**) together with Myc-PGAM5 and HA-WT-Ub or its mutants (HA-K48O-Ub, HA-K63O-Ub or HA-K48R-Ub) were subjected to denaturing IP with the indicated antibodies. $n = 3$ (**e**) independent experiments. Source data are provided as a Source Data file.

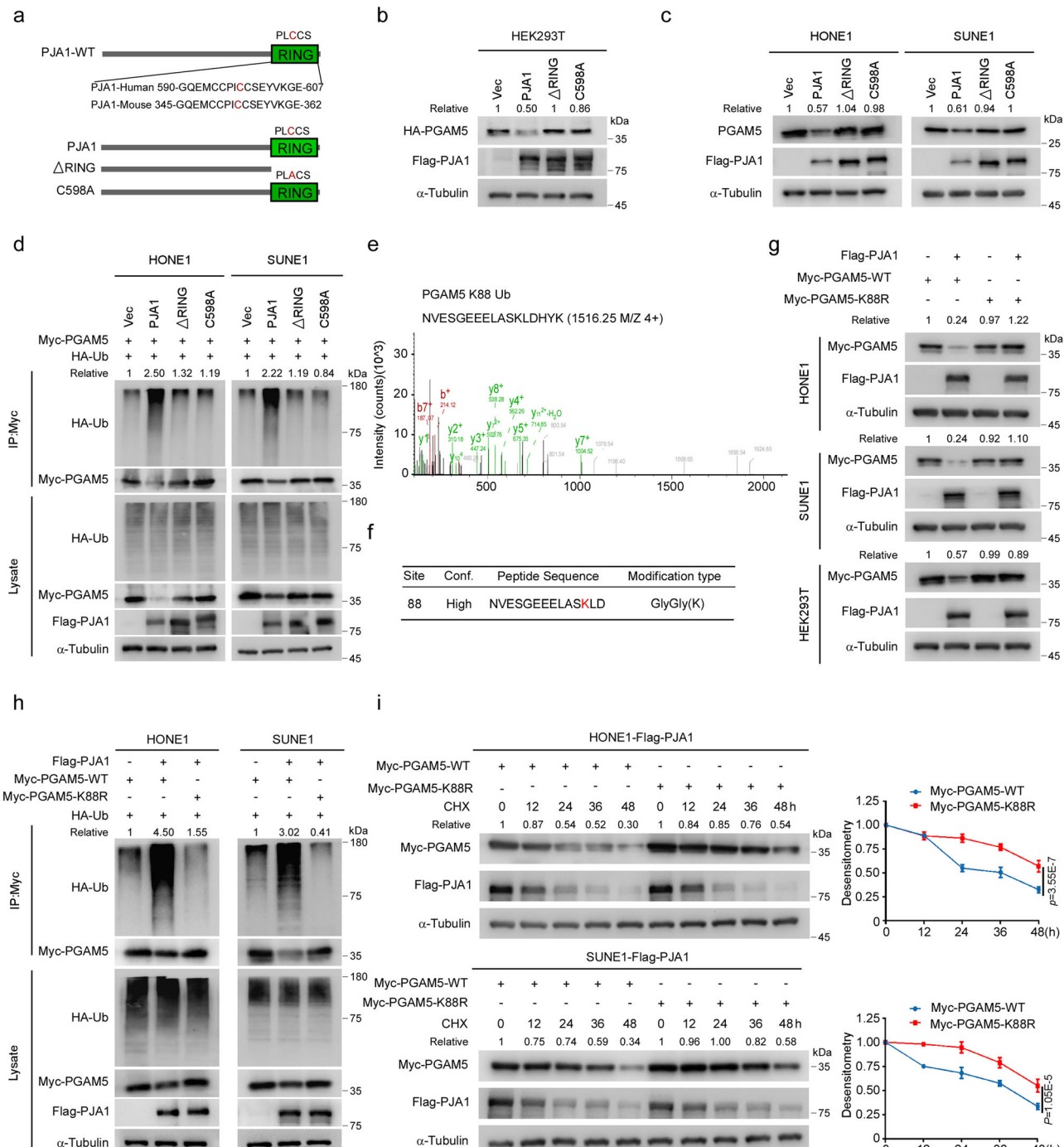

**Fig. 3 | PJA1 catalyses PGAM5 polyubiquitination at K88 depending on its E3 ligase activity. a** Schematic representation of plasmids expressing Flag-tagged wild-type (WT) PJA1, its RING deletion mutant (ΔRING), or its C598A mutant. **b, c** PGAM5 protein levels in HEK293T cells (**b**) and NPC cells (**c**) transfected with the empty vector or the Flag-PJA1 WT, ΔRING or C598A mutant plasmids. **d** NPC cells transfected with the empty vector or the Flag-PJA1 WT, ΔRING or C598A mutant plasmids, together with Myc-PGAM5 and HA-WT-Ub were subjected to denaturing IP with the indicated antibodies. **e, f** Mass spectrometry analysis identified the ubiquitination site K88 in PGAM5. **g** Protein levels of PGAM5 in NPC cells

cotransfected with the empty vector or Flag-PJA1 plasmids, together with the Myc-PGAM5-WT or K88R mutant plasmids. **h** NPC cells were transfected with the empty vector or Flag-PJA1 plasmids, together with the HA-Ub and Myc-PGAM5-WT or the K88R mutant and subjected to denaturing IP with the indicated antibodies. **i** Immunoblot (left) and the corresponding greyscale analysis (right) of PGAM5 expression in NPC cells transfected with the Flag-PJA1 and Myc-PGAM5-WT or the K88R mutant plasmids after CHX treatment (mean ± s.d., two-way ANOVA). N = 3 (**i**) repeats from three independent experiments. Source data are provided as a Source Data file.

aberrancies such as decreased production of ATP and increased production of mitochondrial ROS (mROS)[34–37], we examined the effect of PJA1 knockdown on mitochondrial function, and found that knockdown of PJA1 significantly reduced the production of ATP and increased the level of mROS (Fig. 4h, i). Moreover, mitochondrial

fragmentation has been reported to drive the release of cytochrome c from mitochondria into the cytosol to induce caspase-dependent cell death[38,39], and we found that knockdown of PJA1 significantly promoted the release of cytochrome c (Fig. 4j). These results indicate that PJA1 can suppress docetaxel-induced mitochondrial damage.

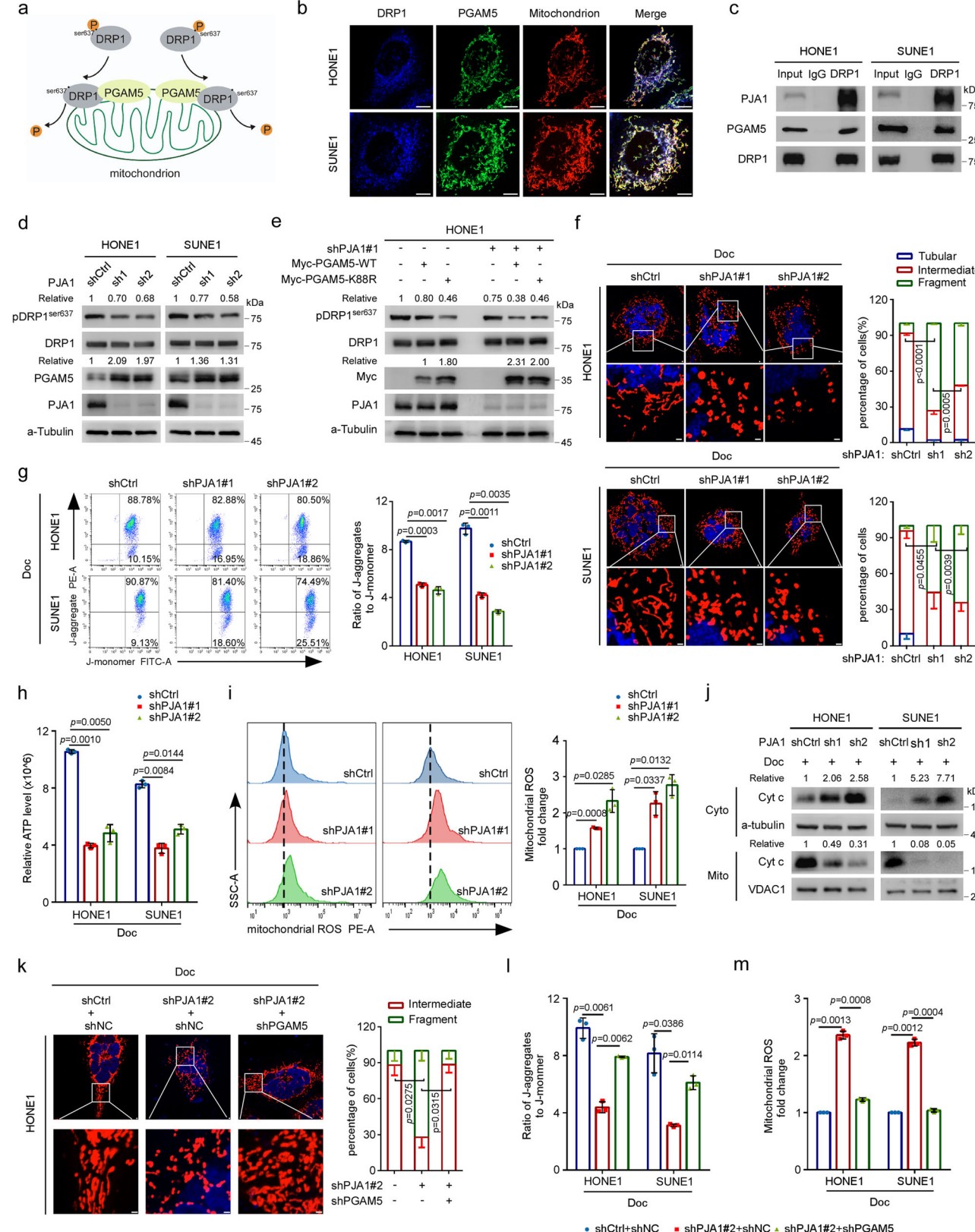

In addition, we observed that knockdown of PGAM5 significantly reversed the effects of PJA1 knockdown on mitochondrial morphology as well as the reduction in the mitochondrial membrane potential and the promotion of mROS production (Fig. 4k–m and Supplementary Fig. 4d–g). Collectively, these findings demonstrate that PJA1 can inhibit docetaxel-induced mitochondrial damage through the PGAM5-DPR1 axis.

## PJA1 degrades PGAM5 to inhibit pyroptosis and promote docetaxel resistance

Next, we investigated whether PJA1 inhibits docetaxel-induced pyroptosis by regulating PGAM5 and found that knockdown of PGAM5 reversed the increase in docetaxel-induced pyroptosis induced by PJA1 knockdown, as evidenced by the pyroptosis-related morphological changes, LDH release, caspase-3/GSDME cleavage and increased

**Fig. 4 | PJA1 inhibits docetaxel-induced mitochondrial damage via the PGAM5-DRP1 axis. a** Diagram showing that PGAM5 recruits DRP1 and dephosphorylates DRP1 at S637 site. **b** IF staining revealed the cellular localisation of endogenous DRP1 (blue), PGAM5 (green) and mitochondria (red). Scale bars, 5 μm. **c** Co-IP with an anti-DRP1 antibody revealed the endogenous associations of DRP1 with PJA1 and PGAM5 in NPC cells. **d, e** Protein levels of total DRP and pDRP1$^{ser637}$ in HONE1 cells transfected with the shCtrl or sh-PJA1s plasmids alone (**d**) or together with the Myc-PGAM5-WT or the K88R mutant plasmids (**e**). **f** Representative fluorescence images of mitochondria in NPC cells transfected with the shCtrl or sh-PJA1s plasmids and exposed to docetaxel (10 nM). Scale bar, 1 μm. More than 50 cells were counted to determine the proportions of tubular and fragmented mitochondria (mean ± s.d., one-way ANOVA). **g–i** The mitochondrial membrane potential (**g**), the production of ATP (**h**) and mROS (**i**) measured by flow cytometry in NPC cells transfected with

the shCtrl or sh-PJA1s plasmids and exposed to Doc (10 nM) (mean ± s.d., one-way ANOVA). **j** Levels of cytochrome c in the cytoplasmic (Cyto) and mitochondrial (Mito) fractions of lysates from NPC cells transfected with the shCtrl or sh-PJA1s plasmids and exposed to Doc (10 nM). **k** Representative images of mitochondria in HONE1 cells transfected with the shCtrl or sh-PJA1 plasmids together with shNC or shPGAM5 plasmids, and exposed to Doc (10 nM). Scale bar, 1 μm. More than 50 cells were counted to determine the proportions of tubular and fragmented mitochondria (mean ± s.d., one-way ANOVA). **l, m** Flow cytometric analysis of mitochondrial membrane potential (**l**) and the production of mROS (**m**) in NPC cells transfected with the shCtrl or sh-PJA1 plasmids together with shNC or shPGAM5 plasmids, and exposed to Doc (10 nM) (mean ± s.d., one-way ANOVA). One-way ANOVA with Dunnett's multiple comparisons test; $n$ (**f–i, k–m**) = 3 independent experiments. Source data are provided as a Source Data file.

---

proportion of annexin V$^+$/PI$^+$ cells (Fig. 5a–d). In addition, the knockdown of PGAM5 almost completely reversed the effects of PJA1 knockdown on docetaxel-induced cell viability and death (Fig. 5e, f).

To further clarify the function of the PJA1-PGAM5 axis in vivo, we established a xenograft model by implanting control SUNE1 cells or SUNE1 cells with stable PJA1 or PJA1-PGAM5 double knockdown into mice, followed by docetaxel injection, and then monitored tumour growth (Fig. 5g). The docetaxel sensitisation effect of PJA1 knockdown was reversed by PGAM5 knockdown (Fig. 5h–j). Moreover, IHC staining showed that DRP1 phosphorylation was decreased in PJA1-knockdown tumours, while PGAM5 knockdown abrogated this effect (Fig. 5k). These data suggest that PJA1 degrades PGAM5 to diminish mitochondrial function, thus inhibiting pyroptosis to promote docetaxel resistance.

### PJA1 attenuates docetaxel-induced antitumour immunity

Since GSDME-mediated pyroptosis promotes the release of DAMPs (e.g., HMGB1 and calreticulin) and then activates DCs and enhances CD8$^+$ T-cell-mediated antitumour immunity[15], we further explored whether PJA1 can regulate antitumour immunity. We performed GSEA based on a public dataset (GSE102349)[40] and found that PJA1-high tumours exhibited weak DC maturation and T-cell activation (Fig. 6a and Supplementary Fig. 5a). Analysis by QUANTISEQ and XCELL algorithms further confirmed that NPC tumours with higher PJA1 expression were infiltrated with fewer DCs and CD8$^+$ T cells than those with lower PJA1 expression (Supplementary Fig. 5b, c), while the infiltration of M1 macrophages and neutrophils did not differ significantly between these tumours (Supplementary Fig. 5d). In addition, the expression level of PJA1 was negatively related to the expression levels of genes related to CD8$^+$ T-cell infiltration and antitumour immune responses, such as CD80, CD86, IFNG, LCK, CD8A and CD69 (Supplementary Fig. 5e, f), suggesting that PJA1 may play a critical role in suppressing DC maturation and CD8$^+$ T-cell-mediated antitumour immunity.

To validate the effect of PJA1 on the infiltration of immune cells, we performed IHC staining and multiplex IF to detect PJA1 expression and CD3$^+$ cells, CD8$^+$ T cells and CD11c$^+$ DCs infiltration in 50 NPC tissue samples with low ($n = 23$) or high ($n = 27$) PJA1 expression (Fig. 6b). Tumours with higher PJA1 expression were infiltrated with fewer CD3$^+$ cells, CD8$^+$ T cells and CD11c$^+$ DCs (Fig. 6c), indicating weaker antitumour immunity in the corresponding NPC patients.

To verify whether PJA1 affects the antitumour immunity activated by docetaxel-induced pyroptosis, we cocultured peripheral monocyte-derived DCs (Mo-DCs) and CD8$^+$ T cells with docetaxel-treated NPC cells. As expected, maturation markers (e.g. HLA-DR, CD83, CD80 and CD86) on Mo-DCs were upregulated when cocultured with PJA1-knockdown NPC cells, while this effect was abolished when cocultured with PJA1-PGAM5 double-knockdown NPC cells (Fig. 6d). Moreover, the percentage of CD69$^+$CD8$^+$ T cells was obviously increased when CD8$^+$ T cells were cocultured with docetaxel-treated PJA1-knockdown NPC cells, and this effect was reversed by PGAM5 knockdown (Fig. 6e).

We further investigated the release of DAMPs in NPC cells treated with docetaxel and found that knockdown of PJA1 increased both the release of HMGB1 from the nucleus into the cytoplasm (Fig. 6f, g and Supplementary Fig. 6a) and the cell surface expression of calreticulin (Fig. 6h and Supplementary Fig. 6b). In addition, the levels of inflammatory factors (e.g. IL-6, IL-1α, IL-1β and TNF-α) that have been reported to be released during pyroptotic cell death[11,41,42] and the levels of CXCL9 and CXCL10, which have been reported to recruit effector T cells into the tumour microenvironment[43], were significantly increased in PJA1-knockdown NPC cells (Supplementary Fig. 6c). The increased concentrations of IL-6, IL-1α and CXCL10 in the supernatant were validated by ELISA (Fig. 6i). Moreover, knockdown of PGAM5 almost fully reversed the increase in the release of these DAMPs in PJA1-knockdown NPC cells (Fig. 6f–i and Supplementary Fig. 6a–c). These findings suggest an essential role of the PJA1-PGAM5 axis in regulating DC maturation and CD8$^+$ T-cell activation.

We further verified the in vivo function of the PJA1-PGAM5 axis in regulating antitumour immunity in an MC38 mouse model (Fig. 6j and Supplementary Fig. 6d). Compared with the control group, the PJA1-knockdown group exhibited reduced tumour growth in terms of size, volume and weight, and the reductions in these parameters were almost reversed by PGAM5 knockdown (Fig. 6k and Supplementary Fig. 6e, f). Flow cytometric analysis showed that PJA1 knockdown significantly increased the infiltration of CD11c$^+$ DCs and CD8$^+$ T cells, as well as the proportion of CD86$^+$CD11c$^+$ tumour-infiltrating DCs (Fig. 6l, m). In addition, the proportions of CD69$^+$ and effector molecule IFN$^+$ and TNF$^+$ cells among tumour-infiltrating CD8$^+$ T cells were increased in tumours with PJA1 knockdown, while knockdown of PGAM5 abolished the above effects (Fig. 6n and Supplementary Fig. 6g). Together, these findings demonstrate that PJA1 attenuates the CD8$^+$ T-cell-mediated antitumour immunity induced by docetaxel through regulation of PGAM5.

### Pharmacological targeting of PJA1 enhances docetaxel sensitivity in NPC

The above results indicate that *PJA1* plays an oncogenic role in NPC and is expected to be an attractive antitumour target to enhance chemosensitivity. A recent study reports that the small molecule inhibitors RTA402 and RTA405 can bind to the RING domain of the PJA1 protein and inhibit its expression[24]. First, we treated HONE1 and SUNE1 cells with RTA402 and RTA405 and found that these NPC cells were sensitive to RTA402 but not RT405 (Supplementary Fig. 7a, b). Through in silico molecular docking simulation, we confirmed that RTA402 and PJA1 exhibited suitable steric complementarity (Fig. 7a). Thus, we further investigated whether RTA402 could be a candidate drug for combination with docetaxel in NPC treatment. Treatment with RTA402 combined with docetaxel reduced NPC cell viability (Supplementary Fig. 7c), indicating that RTA402 increased the docetaxel sensitivity of NPC cells. In addition, treatment with RTA402 combined with docetaxel resulted in downregulated expression of PJA1 and increased expression of PGAM5, caspase-3/GSDME cleavage and LDH

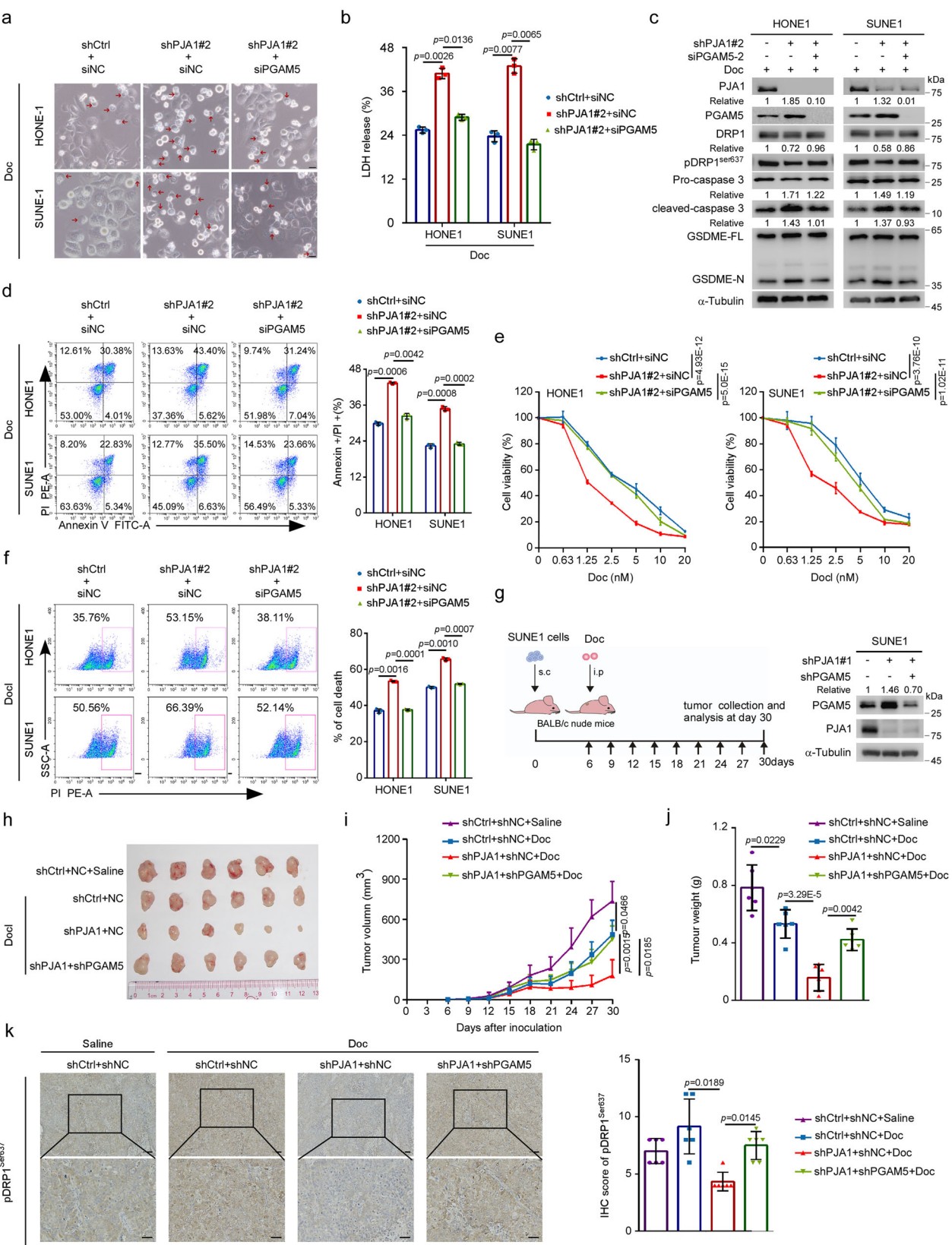

release, which was dependent on PJA1 (Supplementary Fig. 7d–f). To further explore the in vivo function of RTA402, we established a subcutaneous xenograft model. In this model, treatment with RTA402 or docetaxel alone inhibited tumour growth, while combined treatment with RTA402 and docetaxel significantly enhanced the inhibitory effect on tumour growth (Fig. 7b–e). These results confirm that

pharmacological blockade of PJA1 with RTA402 can activate docetaxel-induced pyroptosis to promote the docetaxel sensitivity of NPC.

To determine whether RTA402 enhances antitumour immunity in NPC, we first evaluated the release of DAMPs in NPC cells and found that combined treatment with RTA402 and docetaxel significantly decreased the release of HMGB1 from the nucleus into the cytoplasm,

**Fig. 5 | PJA1 degrades PGAM5 to inhibit pyroptosis and promote docetaxel resistance. a–f** NPC cells were transfected with the shCtrl or sh-PJA1 plasmids in combination with siNC or siPGAM5, and then exposed to docetaxel (10 nM or indicated doses). Representative images of pyroptotic morphology. The red arrow indicates pyroptotic cells. Scale bar, 25 μm. **a** LDH release (**b** mean ± s.d., one-way ANOVA), caspase-3 and GSDME cleavage (**c**), the proportion of annexin V⁺/PI⁺ cells (**d** mean ± s.d., one-way ANOVA), the chemosensitivity (**e** mean ± s.d., two-way ANOVA) and cell death (**f** mean ± s.d., one-way ANOVA) were shown.
**g–k** SUNE1 cells stably transfected with the shCtrl or sh-PJA1 plasmids together with shNC or shPGAM5 plasmids were implanted subcutaneously into the axillae of

BALB/c nude mice to establish a xenograft growth model and exposed to Doc (10 mg/kg) or not (**g**). Macroscopic images (**h**), tumour growth curves (**i** mean (*n* = 6) ± s.d., one-way ANOVA) and the excited tumour weights (**j** mean (*n* = 6) ± s.d., one-way ANOVA) in each group. Representative images of IHC staining (left) and IHC scores (right) for pDRP1$^{Ser673}$ expression in the tumours excised from mice in each group. Scale bars, 50 μm. (**k** mean (*n* = 6) ± s.d., one-way ANOVA). One-way ANOVA with Dunnett's multiple comparisons test; *n* = 4 (**b**), *n* = 3 (**a, d, e, f**) repeats from three independent experiments. Source data are provided as a Source Data file.

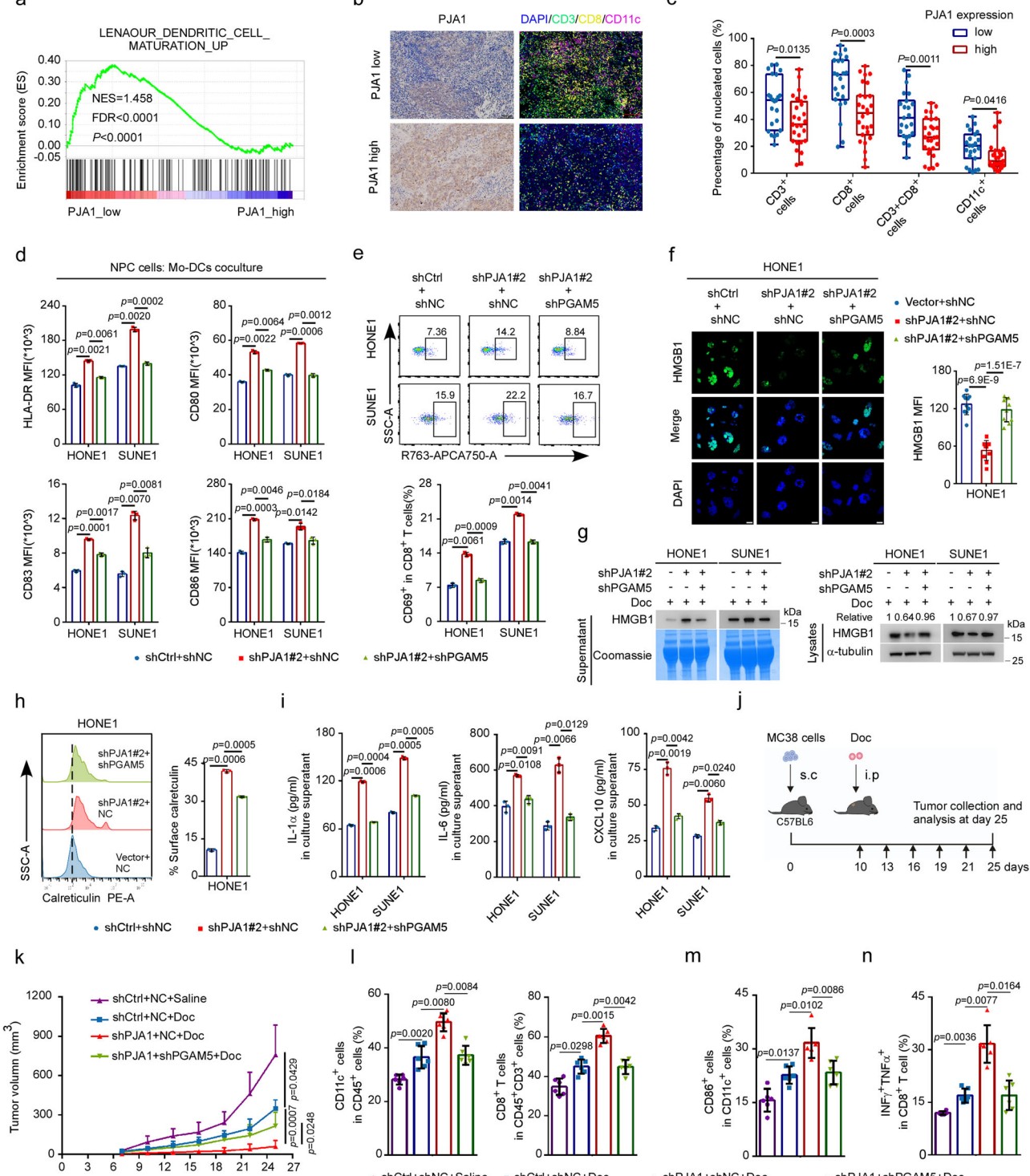

**Fig. 6 | PJA1 attenuates docetaxel-induced antitumour immunity. a** GSEA based on the GSE102349[40] dataset showing gene sets related to DC maturation enriched in PJA1-low NPC tissues (permutation test). **b, c** Representative images (**b**) and quantification results (**c**) of multiplex IF for analysis of infiltrated immune cells in 50 NPC tumours. Scale bar, 100 μm (mean ± s.d. two-tailed unpaired *t*- test). **d–i** NPC cells were treated with Doc (10 nM) for 48 h and were then cocultured with Mo-DCs for another 36 h. the levels of maturation markers on the surface of Mo-DCs were then measured by flow cytometry (**d** mean ± s.d., one-way ANOVA). Doc-treated NPC cells were cocultured with PBMCs for 36 h, and the percentage of CD69+CD8+ T cells was determined by flow cytometry (**e** mean ± s.d., one-way ANOVA). **f** Representative fluorescence images showing the release of HMGB1 from the nucleus into the cytoplasm in HONE1 cells. Quantitative data from ten randomly selected fields per group are shown. Scale bar, 20 μm (mean ± s.d., one-way ANOVA). **g** Levels of HMGB1 in the supernatant and lysates of NPC cells. Coomassie staining was used as a control to verify equal gel loading. **h** The expression level of calreticulin on the surface of HONE1 cells was determined by flow cytometry (mean ± s.d., one-way ANOVA). **i** The concentrations of IL-1α, IL-6 and CXCL10 in the supernatant were measured by ELISA (mean ± s.d., one-way ANOVA). **j–n** MC38 cells were implanted subcutaneously into C57BL/6 mice to establish a xenograft growth model, and these mice were then treated with or without Doc (10 mg/kg) (**j**). Tumour growth curves (**k** mean (*n* = 6) ± s.d., two-way ANOVA), the percentages of CD11c+ DCs among CD45+ cells and the percentages of CD8+ T cells among CD45+CD3+ cells (**l**), the surface expression of CD86 on CD11c+ DCs (**m**), and the percentage of INFγ+TNFα+ T cells among CD8+ T cells (**n**) (mean (*n* = 6) ± s.d., one-way ANOVA) are shown. one-way ANOVA with Dunnett's multiple comparisons test; *n* (**d–f**, **h**, **i**) = 3 independent experiments. Source data are provided as a Source Data file.

increased the cell surface expression of calreticulin (Supplementary Fig. 7g, h), and increased the concentrations of IL-1α, IL-6 and CXCL10 in the supernatant (Supplementary Fig. 7i). We then generated NPC xenografts derived from HONE1 cells in humanised NOD/SCID/IL2rγ null mice and treated these mice with docetaxel alone or in combination with RTA402 (Fig. 7f). Consistent with the results in the first model, RTA402 treatment significantly inhibited tumour growth (Fig. 7g, h and Supplementary Fig. 7j). The proportions of CD45+ cells, CD3+ cells and CD8+ cells were significantly increased in tumours treated with the combination of RTA402 and docetaxel than those treated with docetaxel alone (Fig. 7i). Taken together, these findings indicate that pharmacological targeting of PJA1 with RTA402 enhances docetaxel-induced antitumour immunity and docetaxel sensitivity and that the combination of small molecular inhibitor RTA402 and docetaxel is an effective therapy for NPC.

## Discussion

In this study, we found that PJA1 was upregulated in NPC patients who derived no benefit from TPF IC. PJA1 conferred docetaxel resistance on NPC cells by ubiquitinating and degrading mitochondrial protein PGAM5, further increasing the phosphorylation of DRP1 and decreasing the production of mROS, in turn leading to a suppressive effect on GSDME-mediated pyroptosis and antitumour immunity. In addition, we revealed that RTA402, a pharmacological inhibitor of PJA1, enhanced the docetaxel sensitivity of NPC cells. Clinically, high PJA1 expression in tumours was associated with weak antitumour immunity. High PJA1 expression indicated resistance to TPF IC and a poor clinical prognosis in NPC patients.

Chemoresistance is the main reason for anticancer treatment failure in NPC. Diverse molecular mechanisms, including inactivation of downstream death signalling pathways and changes in the local tumour microenvironment, have been implicated in chemoresistance[44–47]. Zheng et al. reported that the lncRNA TINCR regulates the ACLY-PADI1-MAPK-MMP2/9 axis to facilitate chemoresistance in NPC[48]. Hong et al. demonstrated that circIPO7 facilitates the nuclear translocation of phosphorylated YBX1 and activates the transcription of *FGFR1*, *TNC* and *NTRK1* to promote chemoresistance in NPC[49]. However, the exact mechanism regulating chemoresistance in NPC, especially the modulatory factors in the tumour microenvironment, remains unknown. Here, we found that PJA1 was upregulated in NPC patients who were nonresponsive to TPF IC and determined that it conferred docetaxel resistance via ubiquitination and subsequent degradation of PGAM5 to suppress pyroptosis and antitumour immunity. These findings emphasise the important function of the UPS in chemoresistance in NPC.

PJA1 plays an important role as either an oncogene or a tumour suppressor in several cancers. For example, PJA1 is reported to promote tumorigenesis in glioblastoma by degrading the CIC protein[23]. PJA1 inhibits cell invasion and induces apoptosis by degrading FOXR2 and inactivating the Wnt/β-catenin signalling pathway in lung adenocarcinoma[50]. In addition, numerous proteins have been reported to be targets of PJA1 (e.g. ELF, CIC and pSMAD3)[22–24]. Here, PGAM5 was identified as a direct target of PJA1, a finding that extends the biological function of PJA1 in regulating mitochondrial damage. PGAM5 belongs to the phosphoglycerate mutase family and regulates mitochondrial dynamics and programmed cell death via its atypical serine/threonine phosphatase activity[30,51]. In addition, PGAM5 has been reported to induce chemoresistance to 5-fluorouracil in hepatocellular carcinoma[52]. In this study, we demonstrated that the knockdown of PGAM5 abolished the enhancing effects of PJA1 knockdown on docetaxel-induced pyroptosis, antitumour immunity and NPC chemosensitivity. Moreover, PGAM5 expression was negatively associated with PJA1 expression in tumours in our mouse xenograft model. These results suggest a tumour-suppressive role for PGAM5 in NPC.

Recently, modes of programmed cell death differing from apoptosis have gained increasing attention partially due to the acquisition of apoptosis resistance in tumours[53,54]. Pyroptosis, a newly defined form of programmed cell death, plays an important role in tumour progression and treatment resistance[9,55]. In NPC, treatment with lobaplatin combined with the cIAP1/2 antagonist birinapant is reported to induce pyroptosis by regulating the ripoptosome, ROS release and caspase-3 cleavage[56]. OTUD4-mediated GSDME deubiquitination enhances NPC radiosensitivity by inducing pyroptosis[57]. Here, we found that docetaxel treatment induced GSDME-mediated pyroptosis, an effect that was enhanced by PJA1 knockdown, to facilitate docetaxel sensitivity in NPC. These findings reveal an important role of pyroptosis in chemoresistance in NPC and identify a regulatory role of PJA1 in chemotherapy-induced pyroptosis. In non-small cell lung cancer, docetaxel activates NF-κB signalling to stimulate the release of HMGB1 in a ROS-dependent manner and further recruits CD8+ T cells to promote antitumour immunity[58]. In this study, we demonstrated that the knockdown of PJA1 potentiated docetaxel-induced pyroptosis, increased the release of HMGB1 from NPC cells, and promoted the activation of DCs and CD8+ T cells to induce an antitumour immune response. These findings reveal the role of PJA1 in antitumour immunity and broaden our knowledge of the biological function of PJA1.

Although PJA1 has been reported to be a prognostic indicator in lung adenocarcinoma and hepatocellular carcinoma[24,50], its clinical significance in NPC remains unknown. We found that NPC patients with high PJA1 expression had high risks of death and tumour progression and that these patients could not benefit from TPF IC. This finding reveals a predictive indicator for the individualised therapy of NPC patients. Due to the vital roles of PJA1 in chemoresistance and antitumour immunity in NPC, pharmacological targeting of PJA1 may constitute a therapeutic strategy for NPC. The small molecule inhibitor RTA402 can bind PJA1 protein to inhibit its expression and, in turn, inhibit cell growth in various cancers, such as leukaemia, multiple myeloma, lymphoma, breast cancer, pancreatic cancer and colon cancer[59–63]. Here, we found that RTA402 promoted docetaxel-induced pyroptosis to enhance the chemosensitivity of NPC cells. These

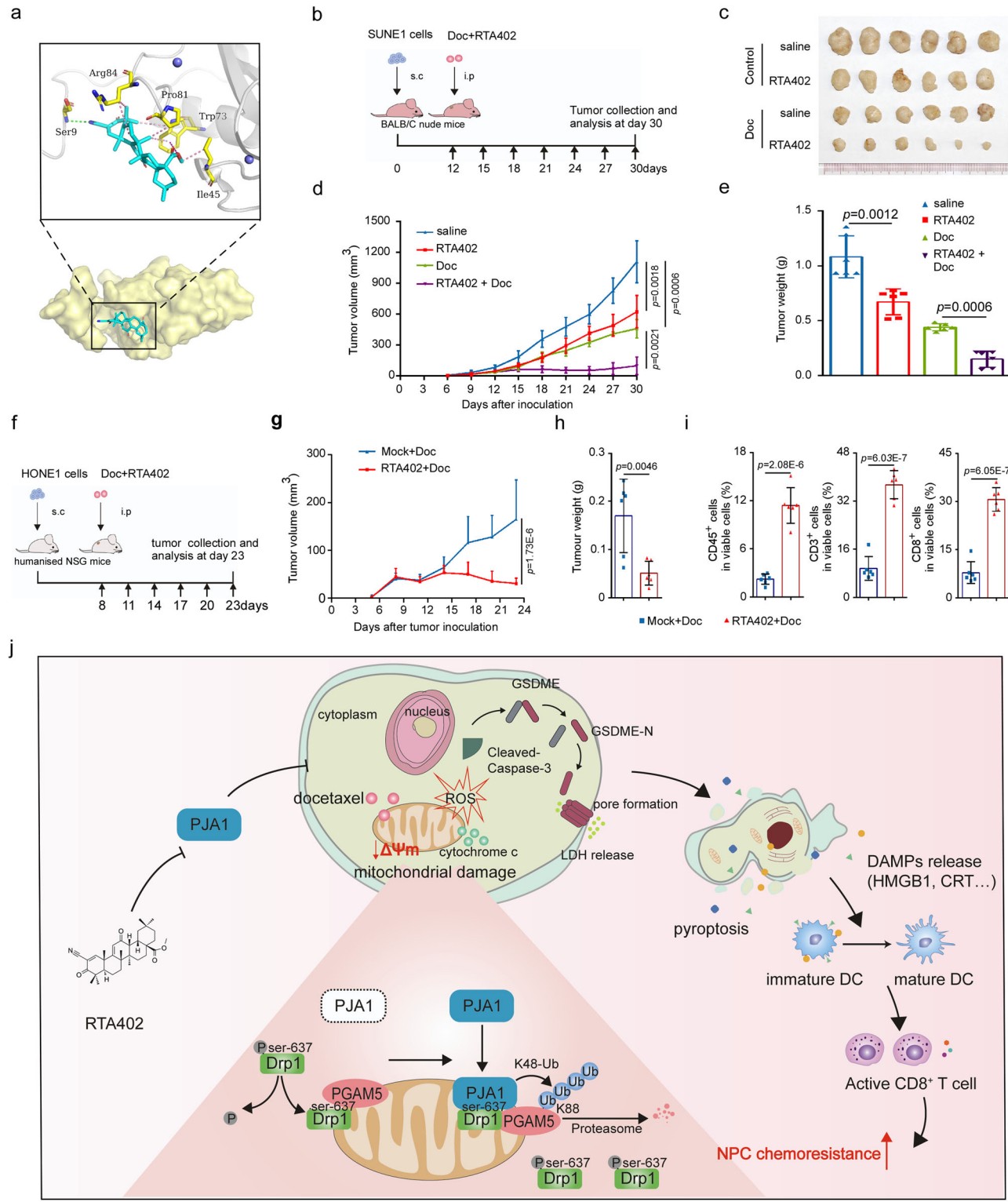

**Fig. 7 | Pharmacological targeting of PJA1 enhances the docetaxel sensitivity of NPC. a** The binding modes of RTA402 to the PJA1 protein were analysed by in silico molecular docking simulation. **b**–**e** SUNE1 cells were implanted subcutaneously into BALB/C nude mice, and these mice were treated with Doc or not (6 mg/kg) together with RTA402 or not (**b**). Macroscopic images (**c**), tumour growth curves (**d** mean ($n = 6$) ± s.d., two-way ANOVA) and the excited tumour weights (**e** mean ($n = 6$) ± s.d., two-tailed unpaired $t$-test) in each group. **f**–**i** HONE1 cells were implanted subcutaneously into humanised NSG mice, and these mice were treated with Doc or not (6 mg/kg) together with RTA402 or not (**f**). Tumour growth curves (**g** mean ($n = 6$) ± s.d., two-way ANOVA), the excited tumour

weights (**h** mean ($n = 6$) ± s.d., two-tailed unpaired $t$-test) and percentages of CD45[+] immune cells, CD3[+] T cells, and CD8[+] T cells (**i** mean ($n = 6$) ± s.d., two-tailed unpaired $t$-test) in tumours from mice in each group. **j** Proposed working model. PJA1 promoted the degradation of the mitochondrial protein PGAM5 by increasing its K48-linked ubiquitination at K88, which further facilitated the phosphorylation of DRP1 at S637 and reduced the production of mROS, resulting in the suppression of GSDME-mediated pyroptosis and induction of the antitumour immune response. PJA1 can be effectively targeted by RTA402 to enhance the docetaxel sensitivity of NPC. Source data are provided as a Source Data file.

observations support the feasibility of combined treatment with RTA402 and docetaxel as a potential therapeutic strategy for NPC. Indeed, a phase 1 clinical trial has been carried out to identify the maximum tolerated dose of RTA402 in patients with solid tumours and lymphomas[64]. More clinical trials are warranted to assess the potential efficacy of the combination of RTA402 and docetaxel in NPC.

In summary, our study reveals a mechanism that E3 ligase PJA1 degrades PGAM5 through K48-linked polyubiquitin chains and facilitates DRP1 phosphorylation, which inhibits GSDME-mediated pyroptosis and antitumour immune response and further promotes docetaxel resistance in NPC. The clinical significance of PJA1 indicates that PJA1 is an independent predictor for poor clinical efficacy of TPF IC in NPC patients and targeting PJA1 by small molecular inhibitor RTA402 enhanced the docetaxel sensitivity of NPC. Our results shed light on the essential role of E3 ligases PJA1 in regulating chemoresistance and provide a therapeutic strategy for NPC based on targeting the UPS.

## Methods

### Ethical statement
This study was approved by the Institutional Ethical Review Boards of Sun Yat-sen University Cancer Centre (G2021-026-01), and the requirement for informed consent was exempted due to the anonymity of the study. All animal experiments were approved by the Experimental Animal Ethics Committee of Sun Yat-sen University Cancer Centre (L025501202108011).

### Clinical specimens
279 paraffin-embedded NPC specimens retrospectively collected between 2009 and 2016 at Sun Yat-sen University Cancer Centre (Guangzhou, China), among which 101 (36.2%) NPC specimens were collected from patients enroled in the TPF IC clinical trial[8]. All patients were restaged according to the 7th edition of the American Joint Committee on Cancer staging manual[65]. All patients received concurrent chemoradiotherapy combined with or without TPF IC. The clinical characteristics of all patients are shown in Supplementary Table 2.

### Cell culture and treatment
The human NPC cell lines HONE1 and SUNE1 were generously supplied by Professor Musheng Zeng from Sun Yat-sen University Cancer Centre and maintained in RPMI-1640 medium (Invitrogen) supplemented with 10% foetal bovine serum (FBS; ExCell Bio). HEK293T cells and MC38 murine colon adenocarcinoma cells were purchased from the American Type Culture Collection (ATCC) and were maintained in DMEM (Invitrogen) supplemented with 10% FBS. All the cells were cultured for less than 2 months, and tested for mycoplasma contamination. None of the cell lines were authenticated. We established a docetaxel-resistant cell line (named S-DR) by exposing SUNE1 cells to increasing concentrations of docetaxel in a stepwise manner over a period of half a year. Transfected NPC cells were treated with 10 μM MG132 (Sigma-Aldrich) or 50 μM CQ (Sigma-Aldrich) for 6 h or with 100 μg/ml CHX (Sigma-Aldrich) for the indicated times (0, 12, 24, 36, and 48 h).

### Plasmids and transfection
The PGAM5 siRNA (siPGAM5) constructs were purchased from RiboBio (China). The sequences of sh-PJA1 and shPGAM5 were designed by shRNA sequence prediction website portals, and the synthesised shRNA oligonucleotides were inserted into a lentiviral vector to construct the PLKO.1-sh-PJA1-1/2 (human), PLKO.1-sh-PJA1 (mouse), PLKO.1-shPGAM5 (human) and PLKO.1-shPGAM5 (mouse) plasmids. The sequences of the shRNAs and siRNAs are shown in Supplementary Table 3. In addition, the PJA1 or PGAM5 coding region was inserted into a vector to construct the phage-CMV-puro-PJA1-Flag and prk-CMV-puro-PGAM5-HA overexpression plasmids. Moreover, the PGAM5 coding sequence was tagged with Myc, Flag or HA and inserted into a vector to construct the pSin-EF2-PGAM5-Myc and pSin-EF2-PGAM5-Flag plasmids. The pCMV-kana-Ub (WT)-HA and pSin-EF2-puro-DRP1-HA plasmids were purchased from Vigene Bioscience (China) and Dahong Biotechnology Corporation (China), and the PRK-HA-Ub (K48O, K63O and K48R) plasmids were generously provided by Professor Bo Zhong from Wuhan University (Wuhan, China).

NPC cells were transfected with the indicated siRNAs or plasmids using Lipofectamine 3000 (Invitrogen), and HEK293T cells were transfected with the indicated plasmids using polyethylenimine (PEI). The transfection efficiency was evaluated by quantitative RT–PCR and western blotting after 24–48 h of transfection.

### Quantitative RT–PCR
Total RNA was isolated by TRIzol reagent (Invitrogen), and cDNA was synthesised by the GoScript Reverse Transcription System (Promega). Quantitative PCR was performed with SYBR Green PCR Master Mix (Invitrogen) or ChamQ SYBR qPCR Master Mix (Vazyme) on a Light-Cycler 480 System (Roche) or a CFX96 Touch sequence detection system (Bio-Rad). GAPDH was used as an internal control, and relative expression was calculated by the $2^{-\Delta\Delta CT}$ method. The primer sequences are shown in Supplementary Table 3.

### Western blot analysis
Western blotting was performed and data were analysed as previously described[66]. Briefly, cells were lysed in RIPA buffer containing phosphatase and protease inhibitors (Roche). Total protein was separated by SDS–PAGE, and transferred to polyvinylidene fluoride membranes (Millipore). The membranes were first blocked in 5% nonfat milk or 5% bovine serum albumin and then incubated with primary antibodies. After incubation with secondary antibodies, the bands of interest were detected with the ChemiDoc MP Imaging System (Bio-Rad). Cell supernatants were harvested and concentrated using Amicon Ultra-15 10 K filters (Sigma-Aldrich). Concentrates of the proteins of interest were analysed by western blotting, and proteins in the gels were stained using Coomassie Brilliant Blue fast staining solution (Fude Biological). Details of the antibodies used are shown in Supplementary Table 4, and the unprocessed scans of the immunoblots are provided in the Source data.

### Chemosensitivity assay
Cells (2000 HONE1 cells or 3000 SUNE1 cells) were seeded into 96-well plates. Beginning 12 h after seeding, the cells were incubated with docetaxel or RTA402 at the indicated concentration for 48 h, and CCK8 reagent was then added for measurement of cell viability according to the manufacturer's instructions.

### IF staining and confocal microscopy/imaging
For examination of pyroptotic morphology, cells were seeded in 6-well plates, incubated for 12 h, and then treated with docetaxel for 36 h. The morphology of pyroptotic cells was evaluated by imaging with a Nikon Eclipse Ti-2 microscope.

For IF staining, cells were sequentially fixed, permeabilized, blocked, and incubated with primary antibodies overnight at 4 °C. After incubation with secondary antibodies, the coverslips were stained with 4′,6-diamidino-2-phenylindole (DAPI; Sigma). Images were acquired using a confocal laser scanning microscope (LSM880 with Fast Airyscan, Carl Zeiss). Multiplex IF was performed by a PANO 7-plex IHC Kit (Panovue) following the manufacturer's instructions. The details of the antibodies used are shown in Supplementary Table 5.

For examination of mitochondrial morphology, cells were stained with MitoTracker Red CMXRos (Beyotime Biotechnology), and the quantification of mitochondrial morphological features was performed by ImageJ software with the MinNA plugin as described

previously[67–70]. Cells were classified into three categories: "tubulated", with most mitochondria maintaining an interconnected and elongated morphology and having a length of more than 10 μm; "intermediate", with mitochondria being mixed with short tubular mitochondria and having a length of less than 10 μm; and "fragmented", in which most mitochondria were punctiform. The analysis was conducted by two researchers blinded to the treatment of these cells.

## LDH release assay

The amount of LDH released into the cell culture supernatant was measured by a CytoTox 96 nonradioactive cytotoxicity assay kit (Promega) following the manufacturer's instructions.

## Flow cytometric analysis

Cells were collected 48 h after docetaxel treatment and washed twice with PBS. For Annexin V/PI analysis, each sample was incubated with the fluorescent dyes Annexin V-FITC (5 μl) and propidium iodide (PI, 5 μl). For mitochondrial membrane potential measurement, each sample was incubated with a JC-1 fluorescent probe following the manufacturer's protocol. For mROS measurement, cells were stained with MitoSOX™ (5 μM). Cells were analysed by a cytoFLEX flow cytometer (Beckman Coulter), and data were analysed with CytExpert 2.2 or Flow Jo software (Tree Star). The gating strategy is provided in Supplementary Fig. 8.

## Mass spectrometry and Co-IP assay

Cells were lysed in IP lysis buffer supplemented with phosphatase and protease inhibitors. Total protein samples were first immunoprecipitated with the indicated antibodies overnight at 4 °C and then incubated with Pierce™ Protein A/G Magnetic Beads (Thermo Scientific) to capture the immunoprecipitated complexes. The collected complexes were washed with IP wash buffer, separated by SDS–PAGE and then stained with a Fast Sliver Stain Kit (Beyotime). Mass spectrometry analysis was carried out by Huijun Biotechnology (China). All ubiquitination assays were conducted under denaturing conditions as described previously[21]. The antibodies used are shown in Supplementary Table 4.

## In vitro coculture assays

Human peripheral blood monocytes (PBMCs) were isolated from healthy donors using CD14 magnetic microbeads (Miltenyi Biotec). Mo-DCs were generated from PBMCs by culture in RPMI-1640 medium supplemented with 10 ng/ml IL-4 (204-IL-010), 25 ng/ml GM-CSF (R&D, 215-GM-010), 10% FBS and 100 U/ml penicillin–streptomycin for 6 days. Then, NPC cells treated with docetaxel for 48 h were cocultured with Mo-DCs or PBMCs for the indicated times. Flow cytometry was used to detect the expression of CD11c, CD80, CD83, HLA-DR, CD86, CD8 and CD69. In brief, Fc receptors (FcRs) on cells were blocked with an FcR-blocking reagent, and the cells were then incubated with the indicated antibodies. For intracellular staining, an Intracellular Fixation & Permeabilization Buffer Set (eBioscience) was used following the manufacturer's instructions. Cells were analysed using a cytoFLEX flow cytometer (Beckman Coulter), and data were analysed with CytExpert 2.2 or Flow Jo software (Tree Star). The antibodies used are shown in Supplementary Table 5, and the gating strategy is provided in Supplementary Fig. 8.

## ELISA

The concentrations of cytokines (IL-1α, IL-6 and CXCL10) released by docetaxel-treated NPC cells were measured with ELISA kits (NeoBioscience) according to the manufacturer's protocols. The cytokine concentrations were considered reliable only when the $R^2$ value of the corresponding standard curve was greater than or equal to 0.99.

## In vivo animal studies

The maximal tumour diameter permitted by our ethics committee is 2 cm, and this was not exceeded at any point. Female BALB/c nude mice and C57BL/6 mice (4–5 weeks old) were purchased from Charles River Laboratories (Zhejiang) and housed in the Animal Experiment Centre of Sun Yat-sen University Cancer Centre. BALB/c nude mice were subcutaneously injected with $7.5 \times 10^5$ sh-PJA1 or vector control SUNE1 cells, and C57BL/6 mice were injected with $1 \times 10^6$ MC38 cells (groups: vec + NC, sh-PJA1 + NC and sh-PJA1 + shPGAM5). The mice were randomly divided into different groups ($n = 6$ per group) when the volume of tumour nodules was ~75 mm³ and were then intraperitoneally injected with docetaxel (10 mg/kg) or saline every 3 days or orally administered RTA402 (7.5 mg/kg every day). The tumour volume was monitored every 3 days. The mice were sacrificed when the maximal tumour diameter was ~1.5 cm, and the tumours were weighed. The tumours were digested into single-cell suspensions to analyse the infiltration of immune cells by flow cytometry. To assess the function of CD8+ T cells, Cell Stimulation Cocktail (plus protein transport inhibitors) (Thermo Scientific) was used to treat the single-cell suspensions for 4 h, and the percentages of IFNγ+TNFα+CD8+ T cells were determined.

Female SPF humanised NSG mice (6–8 weeks old) were purchased from Shanghai Model Organisms Centre, Inc. (Shanghai), and the percentages of human CD45+ cells in the peripheral blood of these mice were determined to be greater than 1% 1 week after tail vein injection of human PBMCs ($5 \times 10^6$). The mice were then subcutaneously injected with $7.5 \times 10^5$ HONE1 cells, randomly divided into two groups ($n = 6$ per group), and intraperitoneally injected with docetaxel or orally gavaged with RTA402 (7.5 mg/kg every day).

## IHC staining

IHC staining was performed and data were analysed as previously described[21,71,72]. In brief, sections were sequentially deparaffinized, rehydrated, preincubated with hydrogen peroxide, subjected to antigen retrieval, blocked, incubated with primary antibodies, labelled with horseradish peroxidase (HRP)-conjugated secondary antibodies, stained with diaminobenzidine (Sigma) and counterstained with haematoxylin. Images were acquired with an AxioVision Rel.4.6 computerised image analysis system (Carl Zeiss). The antibodies used are listed in Supplementary Table 5.

## Bioinformatics analysis

Empirical Bayes (eBayes) statistics in the 'limma package' was used to identify 385 differentially expressed transcripts (empirical fold-change ≥1.5 and eBayes $P$ value <0.05) based on gene expression profiling data (GSE132112)[8]. The Search Tool for the Retrieval of Interacting Genes (STRING, version 11.0; https://www.string-db.org) was used to construct the protein–protein interaction (PPI) network, which used the proteins with a confidence score >0.4. Then, the PPI network data was downloaded and analysed in Cytoscape (version 3.6.0, http://www.cytoscape.org/). The gene sets defined as 'pyroptosis' and 'modulators of TCR and T-cell activation' were computed with GSEA to identify their correlations with PJA1 expression. TIMER2.0 (QUANTISEQ and XCELL algorithm) was used to estimate the tumour immune infiltration between tumours with high (top 50%) and low (bottom 50%) PJA1 expression. The results were visualised with R Studio (version 4.0.3).

Autodock Vina programme was used for molecular docking of ligands with receptors and predicting the binding affinity[73]. Docking was performed to obtain a population of possible conformations and orientations for the ligand RTA402 (CAS: 218600-53-4) at the binding site. All calculations for protein-fixed ligand-flexible docking were done by the Lamarckian Genetic algorithm (LGA) method. The models of the complex were analyzed using Discovery Studio[74]. The interactions of the complex were analyzed using PyMol.

## Statistics and reproducibility

GraphPad Prism version 8 or SPSS Statistics version 25 (IBM) software was used for all statistical analyses. The significance of differences between and among groups was analysed using unpaired two-tailed Student's $t$-test, the chi-square test or Fisher's exact test and two-way or one-way ANOVA with Dunnett's multiple comparisons test as appropriate. Survival curves were plotted using the Kaplan–Meier method, and differences in survival were analysed using the log-rank test. The data were presented as the mean ± s.d. or mean SEM values. The boxplots indicate minimum and maximum (whiskers), 25th and 75th percentiles (bounds of box), and median (centre). $P < 0.05$ was considered statistically significant. All experiments were repeated at least three times.

## Reporting summary

Further information on research design is available in the Nature Portfolio Reporting Summary linked to this article.

## Data availability

Processed data generated in this study are available in the Source Data file. Raw data generated in this study including CCK8, qPCR, western blotting, flow cytometric analysis, immunofluorescence staining, animal experiments and so on, have been deposited in the Research Data Deposit of Sun Yat-sen University Cancer Centre with an accession number RDDB2024700296 (https://www.researchdata.org.cn), which is available upon request for research purpose. The NPC sequencing data used in this study are previously published[8,40] (GSE132112, Gene Expression Omnibus (GEO) repository, https://www.ncbi.nlm.nih.gov/geo/query/acc.cgi?acc=GSE132112), (GSE102349, Gene Expression Omnibus (GEO) repository, https://www.ncbi.nlm.nih.gov/geo/query/acc.cgi?acc=GSE102349). All remaining data can be found in the Main, Supplementary and Source Data files. Source data are provided with this paper.

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

## Acknowledgements

This work was supported by grants from National Natural Science Foundation of China (82172806, N.L.), National Key R&D Programme of China (2021YFA1302100, N.L.), Fundamental Research Funds for the

Central Universities, Sun Yat-sen University (22yklj07, N.L.), and Young Talents Programme of Sun Yat-sen University Cancer Centre (No. YTP-SYSUCC-0010, N.L.).

## Author contributions

N.L., Y.-Q.L., S.-Y.H. and Y.Z. designed the experiments. S.-Y.H., S.G. and M.-L.Y. carried out and analysed the data for most of the experiments. Q.-M.H., J.-Y.W., Y.-L.L. and S.-W.H. collected NPC samples and performed the IHC and multiple IF experiments. Y.Z., S.-W.H., Y.-Q.L. and S.X. helped with the data analyses. J.-Y.L., H.Q., and X.-R.T. helped to perform the bioinformatics analysis. S.-Y.H., Y.Z. N.L. and Y.-Q.L. wrote and revised the manuscript. N.L. and Y.-Q.L. conceived and supervised the study and provided funding and scientific direction. All authors reviewed and discussed the final version of the paper.

## Competing interests

The authors declare no competing interests.
