## [Peer Review File · Nature Communications]

Reviewers' Comments:

Reviewer #1:

Remarks to the Author:

The role of UPS in carcinogenesis is an interesting topic and still a very much growing and unfolding area in oncology. This is an original work that uncovered the key role of UPS in NPC treatment and elucidated the functional mechanism of PJA1, a core factor that confers TPF chemoresistance and regulates anti-tumor immunity. Mechanistically, the authors substantiated that this process is achieved by diminishing pyroptosis through PGAM5-DRP1 axis. The ideas as presented by the authors flow logically and helped organized the content systematically. The data is rich and they have access to a large number of clinical patients for which this information was measured. Overall, this is a study of high-quality and a well written paper. I have few specific comments as follows.

Major comments:

1.Line 102, The authors demonstrated that they observed positive expression of PJA1 in both the nucleus and cytoplasm in NPC tissues. However, from the immunofluorescence in Figure 2C, PJA1 appears to be expressed only in the cytoplasm. What accounts for this difference?

2.What is the workflow process of constructing a drug-resistant cell line S-DR? The authors should indicate this in the Supplementary Figure 2.

3.Figure 4C, The authors performed IP experiments using DRP1 antibodies and found that PJA1 and PGAM5 could be pulled down, suggesting that these three proteins exist as a functional crosstalk complex. However, the author's working model seems to indicate that it is pairwise independent.

4.The authors used molecular docking to identify RTA402, as a PJA1 inhibitor, which significantly inhibited the expression level of PJA1 in vitro. Notably, RTA402 has previously been reported as an IKK inhibitor with strong pro-apoptotic and anti-inflammatory activities. It has been considered as a NRF2 activator and NF- κ B inhibitor. It seems that its pharmacological effects are quite broad. Could the authors explain the underlying mechanism by which RTA402 inhibits PJA1 expression?

5.Some bioinformatics analysis, such as STRING and molecular docking, should be indicated in the method section.

Minor comments:

1.Line 74-78, PJA1 is a new chemotherapy resistance target identified by the authors in this study. I personally feel that this part of the description is more appropriate to appear in the discussion.

2.Please avoid the use of subjective descriptions. For example, line125 'much more', line 248 'almost fully'.

3.Line 146, '20 specified pyroptosis gene sets' is not correct. In fact, only one gene set was used to calculate the pyroptosis index.

4.Figure 6B, The quality of immunohistochemical image of 'PJA1 high' needs to be improved.

5.The expression of positive immune cells should be unified, for example, CD3+CD8+ cells should be written as CD3+CD8+ cells.

6.Italic writing for in vivo or in vitro.

7.Figure 3H, labeling of 'Anti' is needless.

8.Myc tag, Doc, RTA402 should be written in a uniform way in the manuscript. Please confirm.

Reviewer #2:

Remarks to the Author:

The manuscript entitled "The E3 Ligase PJA1 Suppresses Docetaxel-induced Pyroptosis and Antitumour Immunity to Facilitate Chemoresistance in Nasopharyngeal Carcinoma" highlights the role of the E3 ubiquitin ligase PJA1 in ubiquitinating and degrading PGAM5, leading to an increase in DRP1 phosphorylation. Consequently, this process inhibits pyroptosis mediated by GSDME and dampens anti-tumor immunity, thereby conferring resistance of NPC cells to docetaxel.

Specifically, the authors observed an upregulation of PJA1 expression in NPC patients who exhibited ineffectiveness towards docetaxel-cisplatin-5-fluorouracil induction chemotherapy, and this upregulation correlated with adverse clinical outcomes. The authors demonstrated that PJA1 achieves this effect by promoting the K48-linked ubiquitination of PGAM5 at the K88, leading to its degradation. Consequently, this process facilitates the phosphorylation of DRP1 at S637, reduces mitochondrial ROS production, and enhances the activation of dendritic cells DCs and CD8+ T cells, inducing an anti-tumor immune response. Furthermore, the pharmacological inhibitor of PJA1, RTA402, was shown to augment the sensitivity of NPC cells to docetaxel.

The manuscript is well prepared, and the experiments are methodologically sound. I have particular concerns regarding the following aspects.

1. The authors employed techniques including mass spectrometry to analyze and validate the interacting relationship between PJA1 and PGAM5. Of particular interest is whether PJA1 also interacts with other pyroptosis-related proteins?
2. The authors identified that PJA1 can suppress docetaxel-induced pyroptosis and attenuate docetaxel-induced antitumour immunity. Therefore, does docetaxel affect the interaction between PJA1 and PGAM5, as well as the ubiquitination of PGAM5 by PJA1?
3. The authors employed the small molecule inhibitor RTA402 as a pharmacological agent targeting PJA1 and observed suitable steric complementarity between RTA402 and PJA1. Does the activation of docetaxel-induced pyroptosis by RTA402 depend on PJA1?
4. The authors performed IHC staining on 279 paraffin-embedded NPC tissue samples. Is there available data on TPF or docetaxel resistance/sensitivity within these samples? Are there any discernible differences in the protein levels of PJA1 and PGAM5?
5. The schematic diagram in Figure 7J is well-defined. As mentioned in the text, the author notes that PJA1 plays a crucial role as either an oncogene or a tumor suppressor in several cancers. Therefore, it seems more appropriate to confine the proposed working model specifically to NPC, considering the context.

Reviewer #3:

Remarks to the Author:

In the manuscript "The E3 Ligase PJA1 Suppresses Docetaxel-induced Pyroptosis and Antitumour Immunity to Facilitate Chemoresistance in Nasopharyngeal Carcinoma." Huang and colleagues address chemoresistance mechanisms in nasopharyngeal carcinoma (NPC). The authors show that PJA1 is highly expressed in non-responders to docetaxel-based chemotherapy (with poor clinical outcome) and that PJA1 confers resistance to docetaxel-induced pyroptosis. Mechanistically PJA1 induced the ubiquitination and degradation of PGAM5 resulting in the inhibition of pyroptosis and reduced antitumor immunity. The PJA1 small molecule inhibitor RTA402 reestablished sensitivity of NPC to docetaxel in vitro and in vivo.

The presented findings are novel might become clinically interesting.

General comments:

Westernblots are overexposed and not interpretable in many instances
Untreated controls are missing in many cases.

Figure legends and/or figures would benefit from additional information.

Some typos should be addressed in figures and text.

Specific comments:

Untreated controls are missing in Fig 1F and many subsequent figures.

Reduced tumor growth in shPJA1 + saline (Fig1H and I) indicates different growth kinetic of shPJA1 cells (though not visible in Fig1 E). This should be clarified.

Difference shown in Fig1I and Fig7E rather indicate a cytotoxic/cytostatic effect of PJA1 knockdown or inhibition than a sensitization to docetaxel. This needs clarification.

It is not quite clear if experiments for Fig. 1J,K have been done in the presence of docetaxel, if so they should have controls without treatment.

Westernblot in Fig. 1L shows casp3 cleavage which should be accompanied by a reduction of the full length casp3 pool. This is not the case for Hone1.

Error bars in Figure 1 M Sune1 are impressively small. Is this really n=3 independent experiments?

Both cells in Fig 2C seem to be photographed shortly before or right after mitosis. Additional representative images would help to understand the phenotype. In any case visualizing endogenous PJA1 would be much better than overexpression.

All westernblots are overexposed and therefore not easy to interpret.

All westernblots in should be quantified by densitometry.

Phenotypic assessment in Fig4F,K needs to be done by widely available image analysis software.

No or unreadable scale bar in Fig.5 H.

Dear Reviewers,

Thank you very much for giving us the opportunity to improve our study through providing constructive and insightful comments. We have revised the manuscript “**The E3 Ligase PJA1 Suppresses Docetaxel-induced Pyroptosis and Antitumour Immunity to Facilitate Chemoresistance in Nasopharyngeal Carcinoma**” (Ref.: NCOMMS-23-46075) according to your comments. The point-by-point responses to reviewers are provided below. Text has been added or modified, and the changes are highlighted in the Revised Manuscript. We have truly enjoyed the stimulating interactions with the reviewers. We sincerely hope the revised manuscript could publish in *Nature Communications*.

Sincerely yours,

Professor Na Liu, M.D., Ph.D.

State Key Laboratory of Oncology in South China; Guangdong Key Laboratory of Nasopharyngeal Carcinoma Diagnosis and Therapy; Sun Yat-sen University Cancer Center, 651 Dongfeng Road East, Guangzhou 510060, China.

E-mail address: liun1@sysucc.org.cn

ORCID: <https://orcid.org/0000-0001-8654-3636>

The following is the point-by-point response to the reviewers' comments and questions.

Reviewer #1 (Remarks to the Author):

The role of UPS in carcinogenesis is an interesting topic and still a very much growing and unfolding area in oncology. This is an original work that uncovered the key role of UPS in NPC treatment and elucidated the functional mechanism of PJA1, a core factor that confers TPF chemoresistance and regulates anti-tumor immunity. Mechanistically, the authors substantiated that this process is achieved by diminishing pyroptosis through PGAM5-DRP1 axis. The ideas as presented by the authors flow logically and helped organized the content systematically. The data is rich and they have access to a large number of clinical patients for which this information was measured. Overall, this is a study of high-quality and a well written paper. I have few specific comments as follows.

Major comments:

1. Line 102, The authors demonstrated that they observed positive expression of PJA1 in both the nucleus and cytoplasm in NPC tissues. However, from the immunofluorescence in Figure 2C, PJA1 appears to be expressed only in the cytoplasm. What accounts for this difference?

Response:

We thank the reviewer for the insightful comments. To address the reviewer's concerns, we conducted IF staining with endogenous antibody of PJA1 in NPC cells without plasmid transfection. The results showed that PJA1 protein was mainly distributed in the cytoplasm, with a small portion distributed in the nucleus (**Fig. R1**). In addition, the localization of PJA1 protein after overexpression is consistent with that of endogenous PJA1 protein (**Fig. R2**), indicating that the distribution of PJA1 in NPC cells was consistent with that in NPC tissues. As the suggestion by Reviewer #3, we have replaced the IF staining pictures in Fig. 2c, showing undivided NPC cells, to help better understand the phenotype.

Fig. R1: IF staining revealed the cellular localization of PJA1 (purple). Scale bars, 5 μ m.

Fig. R2: IF staining revealed the cellular localization of exogenous FLAG-PJA1 (purple), PGAM5 (green) and mitochondria (red). Scale bars, 5 μ m.

2. What is the workflow process of constructing a drug-resistant cell line S-DR? The authors should indicate this in the Supplementary Figure 2.

Response:

We thank the reviewer for pointing this out. We apologize for not describing the process for constructing docetaxel-resistant cell line S-DR. Briefly, measure the IC₅₀ of NPC parental cells in logarithmic phase to docetaxel. Using 1/10 of IC₅₀ as the initial concentration to stimulate cells for 48 hours, the cells are then digested and cultured in drug free medium for 24 hours and partially cells are cryopreserved. Subsequently, the drug concentration is gradually increased according to the above process, during which the cells that have obtained resistance are cryopreserved. After approximately 6-8 months, NPC cells can stably grow in high concentrations of docetaxel and we can obtain drug-resistant cell lines. We have added a brief description of the process of constructing S-DR cell lines in the main text (**Page 5, line 127**). In addition, we have drawn a flowchart of constructing S-DR (**Fig. R3**) and have added it to the Supplementary Fig. 2e.

Fig. R3: flowchart for constructing a docetaxel resistant cell line.

3. Figure 4C, the authors performed IP experiments using DRP1 antibodies and found that PJA1 and PGAM5 could be pulled down, suggesting that these three proteins exist as a functional crosstalk complex. However, the author's working model seems to indicate that it is pairwise independent.

Response:

We thank the reviewer for kind reminding. The three proteins, DRP1, PJA1 and PGAM5, should exist as a functional crosstalk complex in the working model. We have redrawn the working model to show the correct localization relationship of the three proteins (**Fig. R4**).

Fig. R4: Proposed working model.

4. The authors used molecular docking to identify RTA402, as a PJA1 inhibitor, which significantly inhibited the expression level of PJA1 in vitro. Notably, RTA402 has previously been reported as an IKK inhibitor with strong pro-apoptotic and anti-inflammatory activities. It has been considered as a NRF2 activator and NF-κB inhibitor. It seems that its pharmacological effects are quite broad. Could the authors explain the underlying mechanism by which RTA402 inhibits PJA1 expression?

Response:

Thank you very much for your valuable comments. We have two speculations regarding the reviewer's concerns. First, in a previous study, Chen et al. performed computational molecular docking simulations and docking energy prediction, and found that RTA402 directly bound to the RING domain of PJA1¹. Thus, we speculate that a deubiquitination enzyme can stabilize PJA1 through binding the RING domain of PJA1 since post-translational modifications can regulate protein expression, such as deubiquitination. The enzyme cannot stabilize PJA1 because RTA402 can compete with it and bind to the RING domain of PJA1. Secondly, as mentioned by the reviewer, RTA402 is considered as a NF- κ B inhibitor, while NF- κ B can regulate gene expression at the transcriptional level²⁻⁴. Therefore, we speculate that RTA402 may inhibit the transcription of PJA1 by inhibiting the transcriptional activity of NF- κ B, thereby inhibiting the expression of PJA1. We appreciate the reviewer's insightful feedback, and we acknowledge the merit of delving deeper into the specific mechanism through further exploration.

References:

1. Chen J, et al. Targeting the E3 Ubiquitin Ligase PJA1 Enhances Tumor-Suppressing TGF β Signaling. *Cancer Res.* 80(9):1819-1832. (2020).
2. Feist M, et al. Cooperative STAT/NF- κ B signaling regulates lymphoma metabolic reprogramming and aberrant GOT2 expression. *Nat Commun.* 9(1):1514. (2018).
3. Qie S, et al. ErbB2 activation upregulates glutaminase 1 expression which promotes breast cancer cell proliferation. *J. Cell. Biochem.* 115:498–509. (2014).
4. Zhao Q, et al. Clinicopathological implications of nuclear factor κ B signal pathway activation in diffuse large B-cell lymphoma. *Hum. Pathol.* 46:524–531. (2015).
5. Some bioinformatics analysis, such as STRING and molecular docking, should be indicated in the method section.

Response:

We thank the reviewer for kind reminding. We have added “Bioinformatics analysis” in the Method Section of the revised manuscript (**Pages 24-25, lines 595-612**).

Empirical Bayes (eBayes) statistics in the ‘limma package’ was used to identify 385 differentially expressed transcripts (empirical fold-change ≥ 1.5 and eBayes P value < 0.05) based on gene expression profiling data (GSE132112)¹. The Search Tool for the Retrieval of Interacting Genes (STRING, version 11.0;

<https://www.string-db.org>) was used to construct the protein-protein interaction (PPI) network which used the proteins with confidence score > 0.4. Then, the PPI network data was downloaded and analysed in Cytoscape (version 3.6.0, <http://www.cytoscape.org/>). The gene sets defined as 'pyroptosis' and 'modulators of TCR and T cell activation' were computed with Gene Set Enrichment Analysis (GSEA) to identify their correlations with PJA1 expression. TIMER2.0 (QUANTISEQ and XCELL algorithm) was used to estimate the tumour immune infiltration between tumours with high (top 50%) and low (bottom 50%) PJA1 expression. The results were visualized with R Studio (version 4.0.3).

Autodock Vina program was used for molecular docking of ligands with receptor and predicting the binding affinity². Docking was performed to obtain a population of possible conformations and orientations for the ligand RTA402 (CAS: 218600-53-4) at the binding site. All calculations for protein-fixed ligand-flexible docking were done by the Lamarckian Genetic algorithm (LGA) method. The models of the complex were analyzed using Discovery Studio³. The interactions of complex were analyzed using PyMol.

References:

1. Lei, Y. et al. A gene-expression predictor for efficacy of induction chemotherapy in locoregionally advanced nasopharyngeal carcinoma. *J Natl Cancer Inst.* 113, 471-480 (2021).
2. Trott O, Olson AJ. AutoDock Vina: improving the speed and accuracy of docking with a new scoring function, efficient optimization, and multithreading. *J Comput Chem.* 31:455-461 (2010).
3. Dassault Systèmes BIOVIA, Discovery Studio Modeling Environment. Release 2017; Dassault Systèmes: San Diego, CA, USA, 2016.

Minor comments:

1. Line 74-78, PJA1 is a new chemotherapy resistance target identified by the authors in this study. I personally feel that this part of the description is more appropriate to appear in the discussion.

Response:

Thanks for your valuable comments. In the introduction section, we raised our study's scientific question and our speculation about the biological role of PJA1 in NPC, namely whether PJA1 is a new chemoresistance target for NPC. The description here is to introduce the research topic and make the logic of the article more complete. Specifically, we concluded that PJA1 is a novel chemoresistance target in the discussion section of **(Pages 15, lines 357-364)**, which is consistent with the reviewer's recommendation.

2. Please avoid the use of subjective descriptions. For example, line 125 'much more', line 248 'almost fully'.

Response:

We thank the reviewer for kind reminding. We have removed words related to subjective descriptions in the revised manuscript. They are line 129 "obviously", line 137 "much", line 151 "greatly", line 265 "almost fully", line 303 "obviously", and line 338 "greatly".

3. Line 146, '20 specified pyroptosis gene sets' is not correct. In fact, only one gene set was used to calculate the pyroptosis index.

Response:

We thank the reviewer for kind reminding. We apologized for our negligence. We have adapted the description of "20 specified pyroptosis gene sets" to "one specified pyroptosis gene set consisting of 20 genes" in the revised manuscript (**Page 7, line 158-159**).

4. Figure 6B, the quality of immunohistochemical image of 'PJA1 high' needs to be improved.

Response:

We thank the reviewer for pointing this out. We have improved the quality of the immunohistochemical image of "PJA1 high" and replaced it in the revised Fig. 6b.

5. The expression of positive immune cells should be unified, for example, $CD3+CD8+$ cells should be written as $CD3^+CD8^+$ cells.

Response:

We thank the reviewer for pointing this out. As the reviewer suggests, we have made the expression of positive immune cells consistent (such as $CD3^+CD8^+$) in the revised manuscript.

6. *Italic writing for in vivo or in vitro.*

Response:

We thank the reviewer for kind reminding. As the reviewer suggests, we have written both "*in vivo*" and "*in vitro*" in italics in our revised manuscript (**Page 2, line 39**).

7. Figure 3H, labeling of 'Anti' is needless.

Response:

We thank the reviewer for kind reminding. As the reviewer suggests, we have replaced Anti-Flag with Flag-PJA1 in our revised manuscript.

8. Myc tag, Doc, RTA402 should be written in a uniform way in the manuscript. Please confirm.

Response:

We thank the reviewer for kind reminding. As the reviewer suggests, we have unified the writing of Myc tag, Docetaxel, and RTA402 in our revised manuscript and Figures.

Reviewer #2 (Remarks to the Author):

The manuscript entitled “The E3 Ligase PJA1 Suppresses Docetaxel-induced Pyroptosis and Antitumour Immunity to Facilitate Chemoresistance in Nasopharyngeal Carcinoma” highlights the role of the E3 ubiquitin ligase PJA1 in ubiquitinating and degrading PGAM5, leading to an increase in DRP1 phosphorylation. Consequently, this process inhibits pyroptosis mediated by GSDME and dampens anti-tumor immunity, thereby conferring resistance of NPC cells to docetaxel.

Specifically, the authors observed an upregulation of PJA1 expression in NPC patients who exhibited ineffectiveness towards docetaxel-cisplatin-5-fluorouracil induction chemotherapy, and this upregulation correlated with adverse clinical outcomes. The authors demonstrated that PJA1 achieves this effect by promoting the K48-linked ubiquitination of PGAM5 at the K88, leading to its degradation. Consequently, this process facilitates the phosphorylation of DRP1 at S637, reduces mitochondrial ROS production, and enhances the activation of dendritic cells DCs and CD8+ T cells, inducing an anti-tumor immune response. Furthermore, the pharmacological inhibitor of PJA1, RTA402, was shown to augment the sensitivity of NPC cells to docetaxel.

The manuscript is well prepared, and the experiments are methodologically sound. I have particular concern regarding the following aspects.

1. The authors employed techniques including mass spectrometry to analyze and validate the interacting relationship between PJA1 and PGAM5. Of particular interest is whether PJA1 also interacts with other pyroptosis-related proteins?

Response:

We thank the reviewer for the valuable and insightful suggestions. In our mass spectrometry analysis of SUNE1 cells overexpressing Flag-PJA1, we did not find any pyroptosis-related proteins that interacted with PJA1, such as GSDME, caspase 3, and caspase 9. To address the reviewer’s concern, we conducted co-IP experiments by transfecting Flag-tagged PJA1 plasmids in SUNE1 and HONE1 cells. The results showed that the Flag-PJA1 could interact with the GSDME-FL, but not with pro-caspase 3 and pro-caspase 9 (**Fig. R5**). Notably, knockdown of PJA1 facilitated the cleavage of GSDME, but had no obvious effect on the total protein expression level of GSDME (Fig. 11). Thus, PJA1 regulates NPC cell pyroptosis through binding to PGAM5 and promoting its protein ubiquitination degradation.

Fig. R5: Co-IP with an anti-Flag antibody revealed the association of PJA1 and GSDME-FL, pro-caspase 3 and pro-caspase 9 in NPC cells.

2. The authors identified that PJA1 can suppress docetaxel-induced pyroptosis and attenuate docetaxel-induced antitumour immunity. Therefore, does docetaxel affect the interaction between PJA1 and PGAM5, as well as the ubiquitination of PGAM5 by PJA1?

Response:

We thank the reviewer for the valuable comments. As the reviewer's suggestion, we conducted co-IP experiments with an anti-PJA1 antibody on NPC cells treated with docetaxel or not. The results showed that the endogenous interactions between PJA1 and PGAM5 proteins were increased in NPC cells upon docetaxel treatment (**Fig. R6**). We co-transfected Flag-PJA1, Myc-PGAM5 and HA-Ub plasmids into NPC cells treated with docetaxel or not, and then performed ubiquitination assays. The results indicated that docetaxel treatment promoted the ubiquitination modification of PGAM5 by PJA1 (**Fig. R7**).

To address the reviewer's concern, we have added these results into our revised manuscript (**Page 7, Line 168-169; Page 8, Line 195-196; Supplementary Fig. 3c, i**).

Fig. R6: Co-IP with an anti-PJA1 antibody revealed the endogenous association of PJA1 and PGAM5 in NPC cells exposed to docetaxel(10nM).

Fig. R7: NPC cells transfected with the empty vector or FLAG-PJA1 plasmid together with Myc-PGAM5 and HA-WT-Ub were subjected to denaturing IP with the indicated antibodies.

3. The authors employed the small molecule inhibitor RTA402 as a pharmacological agent targeting PJA1 and observed suitable steric complementarity between RTA402 and PJA1. Does the activation of docetaxel-induced pyroptosis by RTA402 depend on PJA1?

Response:

We thank the reviewer for the valuable suggestions. As the reviewer's suggestion, we conducted western blot assays to test the GSDME cleavage in PJA1-WT and PJA1-KO HONE1 cells exposed to docetaxel and treated with RTA402 or not. The results showed that RTA402 did not affect docetaxel-induced pyroptosis in PJA1-KO cells, but promoted docetaxel-induced pyroptosis in PJA1-WT cells, indicating that the activation of docetaxel-induced pyroptosis by RTA402 depend on PJA1 (**Fig. R8**). We have added the results into our revised manuscript (**Page 14, Line 335; Supplementary Fig. 7f**).

Fig. R8: Representative images of GSDME cleavage in PJA1-WT and PJA1-KO HONE1 cells exposed to docetaxel (10 nM) and treated with RTA402 or not.

4. The authors performed IHC staining on 279 paraffin-embedded NPC tissue samples. Is there available data on TPF or docetaxel resistance/sensitivity within these samples? Are there any discernible differences in the protein levels of PJA1 and PGAM5?

Response:

We thank the reviewer for the valuable suggestions. To determine the clinical significance of PJA1 in NPC patients, we conducted IHC staining with an antibody against PJA1 in a cohort of 279 paraffin-embedded NPC tissues, among which 144 tissues obtained from NPC patients received TPF induced chemotherapy (IC) while another 135 from patients did not receive TPF IC. NPC patients were classified into high and low PJA1 expression groups for Kaplan-Meier survival analysis, and the results showed that patients with high PJA1 expression had significantly worse disease-free survival (DFS), overall survival (OS), and distant metastasis-free survival (DMFS) (Supplementary Fig. 1c-d). Importantly, these results further demonstrated that patients with low PJA1 expression could benefit from TPF IC in terms of better DFS, OS and DMFS, while patients with high PJA1 expression did not benefit from the TPF IC (Fig. 1d, Supplementary Fig. 1f-i). Thus, our data suggest that PJA1 expression can predict the long survival efficacy of TPF IC.

To address the reviewer's concern, we further performed IHC staining with PJA1 and PGAM5 antibodies in 20 paired NPC tumors with response (CR/PR) or non-response (SD/PD) to TPF IC. The results showed that PJA1 protein expression was increase while PGAM5 protein expression was decreased in NPC tumours with non-response to TPF IC than those with response (**Fig. R9**). We have added these results into our revised manuscript (**Page 8, line 182-184, Supplementary Fig. 3g**).

Fig. R9: Representative images of IHC staining and IHC scores for PJA1 and PGAM5 expression in 20 paired NPC tumors with response or nonresponse to TPF IC. Scale bars, 50 μ m (mean (n = 20) \pm s.d., two-tailed unpaired t test).

5. The schematic diagram in Figure 7J is well-defined. As mentioned in the text, the author notes that PJA1 plays a crucial role as either an oncogene or a tumor suppressor in several cancers. Therefore, it seems more appropriate to confine the proposed working model specifically to NPC, considering the context.

Response:

Thanks for your valuable comments. As the reviewer suggests, we have added "NPC chemoresistance" in the working model in Figure 7J of our revised manuscript (**Fig. R4**).

Fig. R4: Proposed working model.

Reviewer #3 (Remarks to the Author):

In the manuscript “The E3 Ligase PJA1 Suppresses Docetaxel-induced Pyroptosis and Antitumour Immunity to Facilitate Chemoresistance in Nasopharyngeal Carcinoma.” Huang and colleagues address chemoresistance mechanisms in nasopharyngeal carcinoma (NPC). The authors show that PJA1 is highly expressed in non-responders to docetaxel-based chemotherapy (with poor clinical outcome) and that PJA1 converts resistance to docetaxel-induced pyroptosis. Mechanistically PJA1 induced the ubiquitination and degradation of PGAM5 resulting in the inhibition of pyroptosis and reduced antitumor immunity. The PJA1 small molecule inhibitor RTA402 reestablished sensitivity of NPC to docetaxel in vitro and in vivo.

The presented findings are novel might become clinically interesting.

General comments:

Western blots are overexposed and not interpretable in many instances

Untreated controls are missing in many cases.

Response:

We thank the reviewer for the constructive suggestions. Following the reviewer’s suggestion, we have added the quantification for all western blots using ImageJ software (Fig. 1-6; Supplementary Fig. 2-4, 6-7), and we also added untreated controls in many experiments (Fig. 1f, g, k, l, m).

1. Figure legends and/or figures would benefit from additional information.

Response:

We thank the reviewer for the constructive suggestions. Following the reviewer’s suggestion, we have added additional information in Figure legends in the revised manuscript.

2. Some typos should be addressed in figures and text.

Response:

We thank the reviewer for pointing this out, and we have corrected the typos in the figures and text of our revised manuscript.

Specific comments:

3. Untreated controls are missing in Fig 1F and many subsequent figures.

Response:

We thank the reviewer for pointing this out. As the reviewer's suggestion, we have added the untreated control. As expected, the cell death rates in "untreated control" group were fewer than 5%, however, the cell death rates in docetaxel-treatment group were greater than 30%, and knockdown of PJA1 significantly promoted cell death (Fig. R10). The results have been replaced in Fig. 1f of our revised manuscript (Page 6, line 130-131, Fig. 1f).

Fig. R10, Flow cytometry analysis of cell death in NPC cells transfected with the shCtrl or sh-PJA1s plasmids and exposed to docetaxel (10 nM) for 48h (mean (n = 3) ± s.d., one-way ANOVA).

4. *Reduced tumor growth in shPJA1 + saline (Fig 1H and I) indicates different growth kinetic of shPJA1 cells (though not visible in Fig 1 E). This should be clarified.*

Response:

We thank the reviewer for the insightful question. In the *in vitro* chemosensitivity experiment of Fig. 1e, our final data was standardized, using the absorbance corresponding to each group's 0nM as the standard. The absorbance of the subsequent concentrations was divided by it to obtain the standardized values for each concentration, which were further used to plot the curve. Therefore, the 0nM of each group was 1, and the inhibitory effect of PJA1 knockdown on growth was not shown. We revealed that knockdown of PJA1 could enhance the docetaxel chemosensitivity of NPC cells (Fig. 1e). Besides, knockdown of PJA1 also inhibited tumour growth and reduced the tumour weight (Fig. 1h, i). Actually, PJA1 expression change simultaneously affected the chemosensitivity and tumor growth in NPC, and this similar double role of several genes has been reported by different research groups¹⁻³.

References:

1. Zheng ZQ, et al. Long Noncoding RNA TINCR-Mediated Regulation of Acetyl-CoA Metabolism Promotes Nasopharyngeal Carcinoma Progression and Chemoresistance. *Cancer Res.* 80(23):5174-5188. (2020).

2. Yuan L, et al. EBV infection-induced GPX4 promotes chemoresistance and tumor progression in nasopharyngeal carcinoma. *Cell Death Differ.* 29(8):1513-1527. (2022).

5. Difference shown in Fig1I and Fig7E rather indicate a cytotoxic/cytostatic effect of PJA1 knockdown or inhibition than a sensitization to docetaxel. This needs clarification.

Response:

Thanks for your valuable comments. In our study, through analysis of a genome-wide mRNA expression profiling dataset (GSE132112), we identified 385 differentially expressed chemosensitivity-related genes between NPC patients with response or nonresponse to TPF IC. We then found that PJA1 is a core gene associated with chemoresistance in NPC, and further demonstrated that knocking down PJA1 could enhance the docetaxel sensitivity of NPC cells through *in vitro* chemosensitivity assay. Thus, PJA1 not only had cytotoxic/cytostatic effect on NPC cells but also increased docetaxel chemosensitivity of NPC cells. Similar to our findings, it has been frequently reported that gene expression changes can affect both the proliferation and chemoresistance at the same time^{1,2}.

References:

1. Shigeta K, et al. IDH2 stabilizes HIF-1 α -induced metabolic reprogramming and promotes chemoresistance in urothelial cancer. *EMBO J.* 42(4): e110620. (2023)

2. Zhang H, et al. CAF secreted miR-522 suppresses ferroptosis and promotes acquired chemo-resistance in gastric cancer. *Mol Cancer.* 19(1):43. (2020)

6. It is not quite clear if experiments for Fig. 1J, K have been done in the presence of docetaxel, if so they should have controls without treatment.

Response:

We thank the reviewer for the constructive suggestion. As the reviewer's suggestion, we have added the untreated controls. The results showed that no morphological features of pyroptosis and obvious LDH release appeared in "untreated control" group, but the morphological features of pyroptosis and the amount of LDH release was induced in docetaxel-treatment group, and knockdown of PJA1 promoted the appearance of

pyroptotic morphological features and LDH release (Fig. R11-12). The results have been replaced in Fig. 1j and k of our revised manuscript (Page 7, line 154-156, Fig. 1j, k).

Fig. R11: NPC cells were transfected with the shCtrl or sh-PJA1 plasmids and exposed to docetaxel (Doc, 10 nM), and the representative images of pyroptotic morphology were shown.

Fig. R12: NPC cells were transfected with the shCtrl or sh-PJA1 plasmids and exposed to docetaxel (Doc, 10 nM), and the LDH release (mean (n = 3) \pm s.d., one-way ANOVA) were tested.

7. Western blot in Fig. 1L shows casp3 cleavage which should be accompanied by a reduction of the full length casp3 pool. This is not the case for Hone1.

Response:

We thank the reviewer for the constructive suggestion. The decrease in full length caspase3 pool was not significant in HONE1 cells, it may be the low proportion of protein undergoing splicing in the full length caspase3. In addition, as the reviewer's suggestion, we have added the untreated controls in Fig. 1l. The results showed that the caspase-3 activation and GSDME cleavage were not detected in untreated group, however, they were detected in docetaxel-treatment group, and knockdown of PJA1 could significantly promote

caspace-3 activation and GSDME cleavage (**Fig. R13**). The results have been replaced in Fig. 11 of our revised manuscript (**Page 7, line 154-156, Fig. 11**).

Fig. R13: NPC cells exposed to docetaxel (10 nM), and then assessed the activation of caspase-3 and GSDME.

8. Error bars in Figure 1 M Sune1 are impressively small. Is this really n=3 independent experiments?

Response:

Thank you for your valuable comment. In this study, we conducted three repeated experiments, but only displaying one technical repetition, which make the error bars relatively small.

9. Both cells in Fig 2C seem to be photographed shortly before or right after mitosis. Additional representative images would help to understand the phenotype. In any case visualizing endogenous PJA1 would be much better than overexpression.

Response:

We thank the reviewer for the insightful comments. To address the reviewer's concerns, we conducted IF staining, and found the colocalization of PJA1, PGAM5, and mitochondria in undivided NPC cells (**Fig. R2**). We have added the new images of IF staining to the revised Figure 2c.

In addition, following the reviewer's suggestion, we conducted IF staining with endogenous antibodies of PJA1 in NPC cells without transfection. The results indicated that PJA1 protein is mainly distributed in the cytoplasm, with a small portion distributed in the nucleus. And the localization of PJA1 protein after overexpression is consistent with that of endogenous PJA1 protein (**Fig. R1**).

Fig. R2: IF staining revealed the cellular localization of exogenous FLAG-PJA1 (purple), PGAM5 (green) and mitochondria (red). Scale bars, 5 μ m.

Fig. R1: IF staining revealed the cellular localization of PJA1 (purple). Scale bars, 5 μ m.

10. All western blots are overexposed and therefore not easy to interpret.

Response:

Thanks for your valuable comments. Following the reviewer's suggestion, we weakened the exposure intensity of the immunoblot bands to easier interpret.

11. All western blots in should be quantified by densitometry.

Response:

We thank the reviewer for pointing this out. Following the reviewer's suggestion, we have added the quantification results for all western blots using ImageJ software¹ as the reviewer suggested.

Reference:

1. Li JY, et al. TRIM21 inhibits irradiation-induced mitochondrial DNA release and impairs antitumour immunity in nasopharyngeal carcinoma tumour models. *Nat Commun.*14(1):865. (2023).

12. Phenotypic assessment in Fig4F, K needs to be done by widely available image analysis software.

Response:

Thanks for your valuable comments. In this study, we assessed mitochondrial length using ImageJ (with MiNA and Fiji plugin) as described previously^{1,2}. Briefly, manually install the MinNA plugin in ImageJ software; open an image and convert it to binary through threshold processing; where the foreground pixel is assigned the maximum value (255) and the background pixel is assigned the minimum possible value (0). Then, the binary image is converted to a skeleton that represents the features in the original image; using the "AnalyzeSkeleton" plugin to classify each pixel of the skeleton and measure the length and number of branches in each skeletonized feature. Finally, mitochondrial morphology is defined³. To address the reviewer's concern, we have added this method in our revised manuscript (**Page 20, line 547**).

References:

1. Khacho M, et al. Mitochondrial Dynamics Impacts Stem Cell Identity and Fate Decisions by Regulating a Nuclear Transcriptional Program. *Cell Stem Cell*.19(2):232-247. (2016).
2. Wu MJ, et al. Epithelial-Mesenchymal Transition Directs Stem Cell Polarity via Regulation of Mitofusin. *Cell Metab*;29(4):993-1002.e6. (2019).
3. Valente AJ, et al. A simple ImageJ macro tool for analyzing mitochondrial network morphology in mammalian cell culture. *Acta Histochem*.119(3):315-326. (2017).

13. No or unreadable scale bar in Fig.5 H.

Response:

We thank the reviewer for pointing this out. We have replaced the image in Fig. 5h to display the scale bar clearly in the figure.

Fig. R14: SUNE1 cells stably transfected with the indicated plasmids were implanted subcutaneously into the axillae of BALB/c nude mice to establish a xenograft model and exposed to docetaxel (10 mg/kg) or not.

Macroscopic images in the excised tumours from each group.

Reviewers' Comments:

Reviewer #1:

None

Reviewer #2:

Remarks to the Author:

The authors responded well to my concerns and this article is suitable for acceptance by NC.

Reviewer #3:

Remarks to the Author:

The authors have responded to many of the questions posed by the reviewers. Nevertheless some points remain open.

A potential growth deficit/toxicity in cells expressing shPJA should be addressed as it might lead to a misinterpretation of all follow up results. A head to head growth curve (non normalised) of shCTRL and ShPJA1 and ShPJA2 cells needs to be shown.

Colocalization of of PJA1 and PGAM5 can not be validated based on images Figure 2C. Flag-PJA1 staining is overexposed/unspecific or simply not colocalizing specifically with PGAM5. This should be revisited and quantified.

All representative western blots are overexposed/saturated and therefore not interpretable nor quantifiable. This casts some doubts on the interpretations by the authors. In any case full membrane scans (of triplicate repeats) should be annexed as a good scientific practice.

Dear Editor and reviewers,

Thank you very much for giving us the opportunity to improve our study through providing constructive and insightful comments. We have revised the manuscript “**PJA1-mediated suppression of pyroptosis as a driver of docetaxel resistance in nasopharyngeal carcinoma**” (Ref.: NCOMMS-23-46075) according to the editor and the reviewers’ comments. The point-by-point responses to reviewers are provided below. Text has been added or modified, and the changes are highlighted in the Revised Manuscript. We have truly enjoyed the stimulating interactions with the reviewers. We sincerely hope the revised manuscript could publish in *Nature Communications*.

Sincerely yours,

Professor Na Liu, M.D., Ph.D.

State Key Laboratory of Oncology in South China; Guangdong Key Laboratory of Nasopharyngeal Carcinoma Diagnosis and Therapy; Sun Yat-sen University Cancer Center, 651 Dongfeng Road East, Guangzhou 510060, China.

E-mail address: liun1@sysucc.org.cn

ORCID: <https://orcid.org/0000-0001-8654-3636>

The following is the point-by-point response to the reviewers' comments and questions.

Reviewer #3 (Remarks to the Author):

The authors have responded to many of the questions posed by the reviewers. Nevertheless some points remain open.

I. A potential growth deficit/toxicity in cells expressing shPJA should be addressed as it might lead to a misinterpretation of all follow up results. A head to head growth curve (non normalised) of shCTRL and ShPJA1 and ShPJA2 cells needs to be shown.

Response:

We thank the reviewer for the constructive suggestions. In the study, our results showed that knocking down PJA1 could enhance the docetaxel sensitivity of NPC cells through in vitro chemosensitivity assay. We also found that knocking down PJA1 inhibited the growth of NPC cells in vivo animal experiments, and the effect was stronger under treatment with docetaxel, indicating that PJA1 not only had cytotoxic/cytostatic effect on NPC cells but also increased docetaxel chemosensitivity of NPC cells. Consistent with our findings, many studies have shown that gene expression changes can affect both proliferation and chemoresistance at the same time¹⁻³. However, the inhibitory effect of PJA1 knockdown on growth was not shown in vitro chemosensitivity assay because the final data in Figure 1e was standardized, using the absorbance corresponding to each group's 0nM as the standard. In our previous experiments, we conducted both chemosensitivity and proliferation experiments, but we only presented chemosensitivity experiments' results and did not show proliferation experiments' results in the manuscript. Thus, to address the reviewer's concern, we showed the head to head growth curve (non normalised) of shCtrl and ShPJA1 and ShPJA2 cells below (**Fig. R1**), and the results indicated that knocking down PJA1 could inhibit NPC cells growth. In addition, we have added these results into our revised manuscript (Page 6, Line 126-127; Supplementary Fig. 2e).

Fig. R1: CCK-8 assays in SUNE-1 and HONE-1 cells transfected with shCtrl and ShPJA1 and ShPJA2.

1.Sun Y, et al. METTL3 promotes chemoresistance in small cell lung cancer by inducing mitophagy. J Exp Clin Cancer Res. 242(1):65. (2023).

2. Zhang H, et al. CAF secreted miR-522 suppresses ferroptosis and promotes acquired chemo-resistance in gastric cancer. Mol Cancer. 19(1):43. (2020)

3.Shigeta K, et al. IDH2 stabilizes HIF-1 α -induced metabolic reprogramming and promotes chemoresistance in urothelial cancer. EMBO J. 42(4): e110620. (2023)

2.Colocalization of PJA1 and PGAM5 can not be validated based on images Figure 2C. Flag-PJA1 staining is overexposed/unspecific or simply not colocalizing specifically with PGAM5. This should be revisited and quantified .

Response:

Thanks for your valuable comments. To confirm the colocalization of PJA1 and PGAM5, we have redone the immunofluorescence experiment and quantified the distribution intensity (Pearson correlation coefficient) and the overlap degree (Manders overlap coefficient) of the two proteins (**Fig. R2**). The quantitative data showed that the Manders overlap coefficients of PJA1 and PGAM5 protein in NPC cells were 0.96554 and 0.99514, respectively, and the Pearson coefficients are 0.55347 and 0.4936. These results indicated that there is co-localization between Flag-PJA1 and PGAM5 proteins in NPC cells. To address the reviewer’s concern, we have replaced the IF images in Figure 2c. with **Fig. R3**.

Fig. R2: IF staining revealed the cellular localization of exogenous FLAG-PJA1 (purple) and PGAM5 (green). Right: representative image; left: quantitative data. Scale bars, 5 μ m. .

Fig. R3: IF staining revealed the cellular localization of exogenous Flag-PJA1 (purple), endogenous PGAM5 (green) and mitochondria (red) in NPC cells. Scale bars, 5 μ m.

3. *All representative western blots are overexposed/saturated and therefore not interpretable nor quantifiable. This casts some doubts on the interpretations by the authors. In any case full membrane scans (of triplicate repeats) should be annexed as a good scientific practice.*

Response:

We thank the reviewer for pointing this out. As the reviewer's suggestion, we have adjusted the background of all western blotting bands to make it not overexposed or oversaturated, which didn't affect our experimental results and kept the experimental conclusions consistent with the previous ones. It should be noted that our western blotting bands were exposed using the Bio-Rad Imaging System instead of traditional X-ray film method, so the exposure intensity of western blotting bands can be adjusted through adjusting background intensity using Image Lab software. We cut the PVDF membrane into small bands according to the proteins' size, and incubated them with antibodies. Therefore, our original images were not the entire membranes, but small bands. All of our western blotting experiments were conducted at least 3 times, and here we only showed the original images of one western blotting experiment.